# C-LLAMA 1.0: a traceable model for food, agriculture, and land-use

Thomas S. Ball[1], Naomi E. Vaughan[1,2], Thomas W. Powell[3], Andrew Lovett[1] & Timothy M. Lenton[3].

[1]School of Environmental Sciences, University of East Anglia, Norwich, NR4 7TJ, UK

[2]Tyndall Centre for Climate Change Research, University of East Anglia, Norwich, NR4 7TJ, UK

[3]Global Systems Institute, University of Exeter, Exeter, EX4 4PY, UK

*Correspondence to*: t.ball@uea.ac.uk

**Abstract.** We present C-LLAMA 1.0 (Country-level Land Availability Model for Agriculture), a statistical-empirical model of the global food and agriculture system. C-LLAMA uses simplistic and highly traceable methods to provide an open and transparent approach to modelling the sensitivity of future agricultural land-use to drivers such as diet, crop yields and food-system efficiency. C-LLAMA uses publicly available FAOSTAT food supply, food production, and crop yield data to make linear projections of diet, food system and agricultural efficiencies, and land-use at a national level, aiming to capture aspects of food systems in both developing and developed nations. In this paper we describe the structure and processes within the model, outline an anchor scenario, and perform sensitivity analyses of key components. The models land-use output behaves as anticipated during sensitivity tests and under a scenario with a prescribed reduction in animal product consumption, in which land-use for agriculture is reduced by 1.8 Gha in 2050 when compared with the anchor scenario.

## 1 Introduction

Land-use plays a critical role in achieving Paris Agreement temperature goals. Favoured climate change mitigation strategies such as biomass energy with carbon capture and storage (BECCS) and afforestation rely heavily on widespread land-use change to achieve the necessary scales to be effective (Gough et al., 2018; Roe et al., 2019; Rogeli et al., 2018; Vaughan et al., 2018). However, a range of interlinked factors may jeopardise the sustainable deployment of these mitigation strategies; these include carbon leakage, ecosystem services and biodiversity, and the need for land to support human livelihood and food supply (Arneth et al., 2019). With growing global populations and wealth there are also increasing demands for food quantity and diversity, placing additional pressure on the agricultural system and corresponding land use to meet the demand (Allen et al., 2018; Alexander et al., 2016).

Integrated assessment models (IAMs) make comprehensive projections of future scenarios by coupling economics and land-use with simple carbon cycle and climate models. These models are driven by macro-economics, using a combination of dynamic and static input factors to project future scenarios and are the basis of the Paris Agreement warming targets (UNFCCC, 2015). Most IAMs deal with land use, although there are some exceptions. IAMs are well suited to holistic modelling of future scenarios, especially with the objective of informing policy. They are able to draw together a wide variety of physical, social, and economic processes to produce informed estimates of future scenarios; their mechanisms are well

documented and many are open source (Havlík et al., 2014; Popp et al., 2014; Calvin et al., 2013; Fujimori et al., 2012; Vuuren et al., 2011). However, from their complexity arises an element of nebulousness, they are not able to undertake more detailed analysis of more specific aspects independent of the whole. Despite the broad applicability of IAMs, there remains a need for

models of reduced complexity; they are able to undertake more specific analyses of components that more complex models like IAMs are unable to represent individually. There are significant strengths and weakness to both approaches and they are best used in conjunction with one another, somewhat analogous to reduced-complexity climate models and their general-circulation counterparts (R. J. Nicholls et al., 2020; Sarofim et al., 2021)

FALAFEL (Flux Assessment of Linked Agricultural Food production, Energy potentials & Land-use change) is a global-level

model, using linear projections of global food supply, agricultural efficiencies, and yields to produce trajectories for land-use, carbon capture and energy to 2050 (Powell, 2015; Powell & Lenton, 2012). C-LLAMA (Country-Level Land Availability Model for Agriculture) is the successor to FALAFEL; it is based on the same principles and processes as FALAFEL but disaggregated to the country level. It produces a land-use trajectory to 2050 for each food commodity and commodity group within a country. Where a global model cannot represent the differences between the food systems in a highly developed

country and a developing one, C-LLAMA is able to. This is the primary advantage of moving to a country-level model: it allows for the exploration of the drivers of land-availability in the across a variety of food systems. C-LLAMA is built in Python (Van Rossum and Drake Jr, 1995), unlike FALAFEL which is built in Microsoft Excel. The purpose of the model is to be transparent and easily traceable, as such the model code is open-source and uses only publicly available data as it's inputs. C-LLAMA is situated at the opposite end of the modelling spectrum to IAMs; taking a bottom-up approach to modelling future

land availability; beginning with food supply, then projecting food demand and production forward. In a similar approach to that of FALAFEL, Bijl et al. (2017) consider the relationships between income and dietary patterns to model long-term food demand, but halt at the crop demand stage. C-LLAMA has no economic considerations but models the full range of the food-system from the consumer to the production of crops and animal products. Where FALAFEL and Bijl et al. model the food-system at a global and regional level, C-LLAMA operates at a national level.

**2 Model overview**

C-LLAMA is a statistical-empirical model that uses data from the FAOSTAT database as its primary input (Food and Agriculture Organisation of the United Nations, 2017). These datasets contain food supply and production data, with the food-balance sheets used containing data from 1961 to 2013, and all other datasets (such as land-use and production) running from 1961 to 2017. All data is at a country-level. C-LLAMA models the same timespan as FALAFEL: from 2017 to 2050. Many

of the processes in the model are the same as those in the FALAFEL but operate at a country level as opposed to being globally aggregated. An overview of the structure of C-LLAMA is given in Fig. 1. A list of all modules responsible for model processes in C-LLAMA, grouped into model sections, can be found in Appendix A.

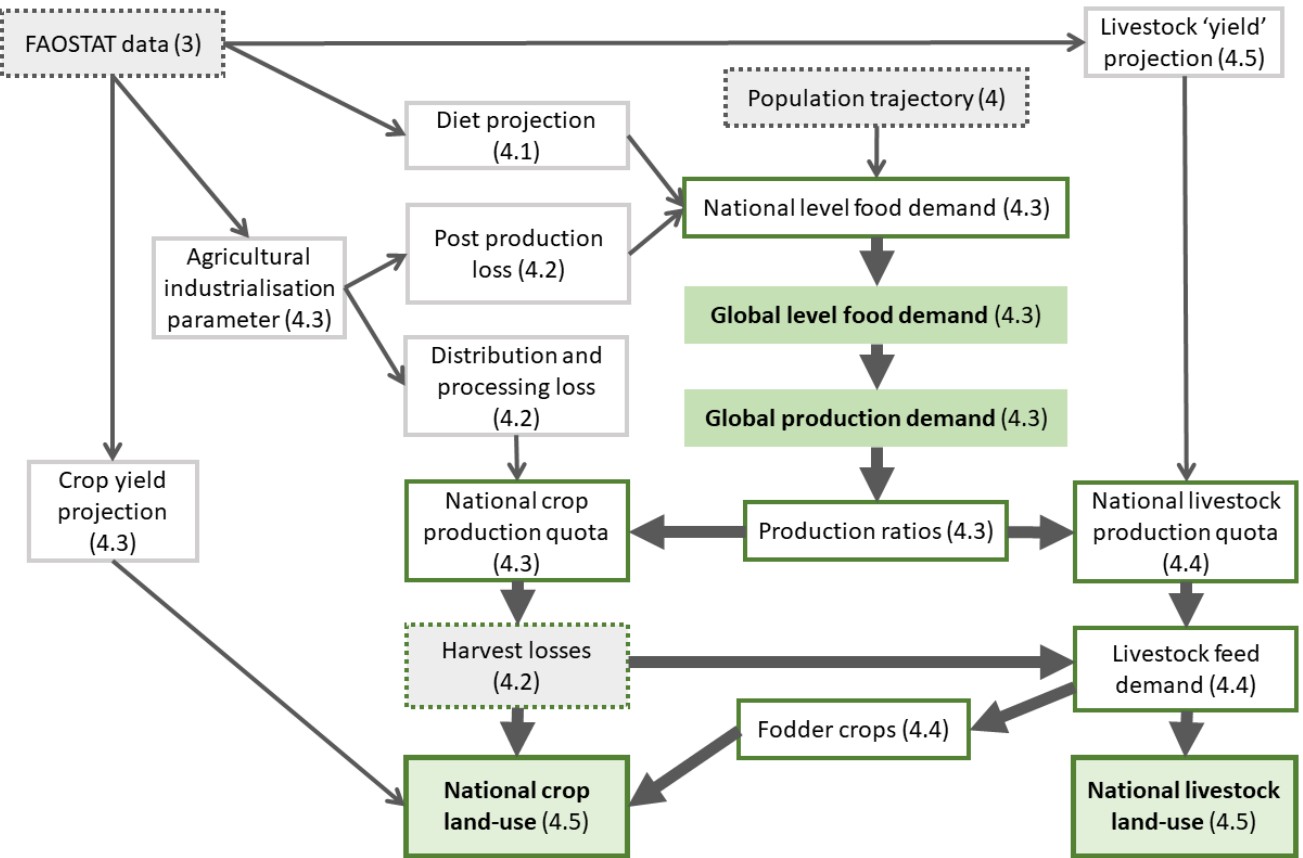

**Figure 1: Overview of C-LLAMA Model structure and flow, with relevant section numbers within the paper indicated in parentheses. Boxes with a dotted border are external datasets while a solid border represents values calculated in C-LLAMA. Thick arrows represent a flow of mass or energy, thin arrows represent the contributing trajectories or factors. Boxes outlined in green are core processes. Boxes shaded in green are globally summed quantities. National crop land-use and livestock land-use are shaded and outlined in green, to highlight them as the primary output of the model. Not all model processes and connections are depicted, this diagram gives a general overview of C-LLAMA.**

The model operates across five continents: Africa, the Americas, Asia, Europe and Oceania, C-LLAMA then splits these into further subcontinental regions (for example, the Americas are split into N. America, S. America, Central America and the Caribbean), most of which contain several countries or states. The model is structured into the following four spatial aggregations: global, continent, region, and country, aligning with the United Nations Statistics Division (UNSD). The structuring of the model into these spatial aggregations allows modifications to be targeted at specific levels. All model processes operate at the country level, with the exception of total global level food demand and global production demand, which are globally aggregated. Food production is then allocated at the country-level.

Global food production and demand is dominated by a small handful of countries. For example: Brazil, the USA and Argentina together accounted for 52% of production by mass of crops used for food in 2017. Of the 162 countries in the FAOSTAT data (that produced food in 2017), the 100 most food-productive countries account for 99.7% of the total production mass. The remaining 62 countries account for only 0.3% of the total food production. Countries whose food production mass in 2017 equates to less than 0.01% of the 2017 global total *and* whose agricultural land-area is less than 34,000 hectares are excluded from the model processes. Figures illustrating this can be found in appendix A. This is done in C-LLAMA for two reasons. The first is to reduce unnecessary model run-time and development complexity. The second reason is that many of these countries have reduced data quality and availability due to their size. Often the data is discontinuous, most commonly due to changes in reporting or assessment. This can lead to unrealistic behaviour when making projections of the data as C-LLAMA does.

There are a small number of countries not included in the model processes because no food balance data for them is available from the FAOSTAT database. The reason for this in most cases is a recent history of political instability or conflict, which suggests that motivating land-based climate mitigation action in these regions may be difficult (World Bank, 2020). Notable for their large land areas, Libya, Sudan, Somalia, and the Democratic Republic of the Congo in Africa (DRC), and Papua New Guinea in Oceania are not included in the dataset, a total land area of 500 Mha. Despite their large land areas, Libya, the DRC, and Papua New Guinea have a small amount of agricultural land for their size at less than 10%, and as low as 2% in the case of Papua New Guinea. Sudan has 40% agricultural land coverage and Somalia has 70%.

## 3 Model components

### 3.0.1 Population

C-LLAMA uses population trajectories from the shared socio-economic pathways (SSP) database, available as 5-yearly population values for each country. SSP2 is a middle of the road scenario with corresponding population projection based on medium values for fertility, mortality, education and migration (KC and Lutz, 2017). The SSP2 population projection is used as a default but any population projection data can be applied. The population data is interpolated linearly to produce a yearly population trajectory to 2050.

### 3.1 Food supply

We define food supply for a given country to be the mean number of kilocalories available per capita per day in a given year. This includes any post-production food waste; some food reaches consumers but is never eaten, either commercially or as domestic waste. The proportion of food wasted in this way is as high as 30% in most developed countries (Alexander et al., 2017).

FAOSTAT food balance sheets contain food supply data disaggregated into different food commodities (Food and Agriculture Organization of the United Nations, 1997). C-LLAMA uses this data to produce a projected food demand for each country.

First, a regression line is calculated for the total food supply for a given country in the period 1961 to 2013, which is then used to calculate a projected food supply value for the year 2050. A linear projection is made for each country from their current total food supply to the projected 2050 total food supply, using the following equation:

$$F(n) = F_0 + \frac{n-n_0}{n_{\text{target}}-n_0}\left(F_{\text{target}} - F_0\right) \tag{1}$$

where $F_n$ is the total food supply in year $n$, $F_{\text{target}}$ is the projected 2050 total food supply per capita, $F_0$ is the mean of the most recent five years of historical food supply data. $n_0$ and $n_{\text{target}}$ are the start and end years of the projection, 2013 and 2050.

Secondly, a linear regression is used to make a projection for the calorie supply from each of the food groups animal products, vegetal products, and aquatic products. Regression lines with a p-value greater than 0.05 are discounted (this threshold value can be changed), instead fixing the projection at the mean value of the most recent five years of data. These projections are then converted into fractions. The proportion of food supply $(P)$ made up by group $i$ in year $n$, is given by

$$P_i(n) = \frac{a_i n + b_i}{\sum_{g \in G}\left(a_g n + b_g\right)}, \tag{2}$$

where $a$ and $b$ are the gradient and intercept of the regression line for that group and $G$ is the set of groups: animal, vegetal and aquatic products.

Third, another linear regression is used to project the relative proportions of individual food commodities within the three food groups. Key food commodities are represented individually, for example wheat, maize and rice in the vegetal product group, and bovine meat and poultry meat in the animal product group. Other commodities are represented in groups, for example 'cereals – other' contains all cereals that are not singled out as key commodities, while the 'luxuries' group contains all tea and coffee. Aquatic products are not the focus of the model as they have minimal to no land requirements during their production; thus they are placed in a single group. Hence, in C-LLAMA, aquatic products simply offset some of the calorific demand from the other food groups. Where possible, C-LLAMA uses vegetal product groups defined in FAOSTAT data. A full list of food commodities and groupings can be found in Appendix B. The commodities within a group are then converted into ratios, so the proportional calorific contribution of commodity $j$ to its umbrella food group $i$ in year $n$ is

$$P_j(n) = \frac{a_j n + b_j}{\sum_{c \in C}\left(a_c n + b_c\right)} \tag{3}$$

where $a$ and $b$ are the gradient and intercept respectively of the regression line for that commodity and $C$ is the set of commodities within the group, for example if $j$ is wheat then $C$ would be all vegetal products. The structure of the projected food supply is then as follows: the total calorie projection is apportioned to each of the food groups by their projected ratios, which are in turn apportioned to the projected commodity ratios. Hence by combining equations 1, 2 and 3, the number of calories contributed to the mean daily food supply per capita by commodity $j$ (of group $i$) is

$$E_j(n) = F(n) * P_i(n) * P_j(n), \tag{4}$$

where all symbols have their previously defined meanings. This approach facilitates the tuning of dietary scenarios by modifying the growth rate of the animal product group or dairy commodities to simulate increases in vegetarianism or veganism.

## 3.2 Food system efficiency

### 3.2.1 Food system efficiency parameter

There is significant variation in food system efficiency, both at different stages and between developed and developing food systems. To reflect this in C-LLAMA, a parameter was developed to assign areas an appropriate degree of efficiency at each stage of the food system and in the model processes. The requirements of the system are the following:

1. Allow the food system efficiency of states to improve as the model progresses.
2. Limit improvement to a realistic maximum.
3. Be representative of most real-world cases. Outliers are inevitable but significant contributors of food demand or food production to the global food system should be captured well.

A highly developed nation in which the majority of farming practices are heavily industrialised with high levels of efficiency should have a score of greater than 1.0 whilst a less developed country in which the majority of people are fed through subsistence farming should score lower than 0.5. A metric such as GDP per capita is not suitable, because a state with extreme income equality could score highly when in actuality the majority of inhabitants rely on subsistence agriculture. Other metrics such as irrigation, fertiliser use and agricultural machinery density were all considered. However, each of these metrics can be skewed by climate, crop types and traditional practices. As such these are also not always reflective of the relative agricultural efficiency of an area.

A parameter was developed based on the yearly mean of daily food energy consumption per capita. This is a self-moderating quantity: unlike GDP there is a maximum realistic value that this can take regardless of economic disparity, so the mean cannot be skewed by extreme cases. The equation for the food system efficiency parameter $X$ for a country $a$ in year $n$ is

$$X_{a,n} = \frac{F_{a,n}}{F_{\text{target}}*0.7} - \frac{0.5}{0.7} \tag{5}$$

where $F$ is the country's total food supply in year $n$. $F_{\text{target}}$ is an idealised food supply, defined as 2500 kcal per capita per day with an additional 30% lost to post-production food waste (see Table 1). This is representative of the food supply in the majority of highly developed regions (N. America, Europe, and Australia and New Zealand) (Kearney, 2010; United Nations Environment Programme, 2021). . Using the ratio of food supply to an idealised food supply generates values in the approximate range 0.5 to 1.2 for the year 2013. The values 0.5 and 0.7 scale the metric to produce values for $X_n$ in the range 0.0 and 1.0.

This parameter is then projected forward with a simple linear projection to 2050 for use in the model processes. In the very few cases where the projection prescribes a decline in food system efficiency, the parameter is halted at the most recent historic value. In the majority of cases this parameter reasonably depicts the position of a country along a scale between complete subsistence agriculture to an industrialised nation with developed infrastructure. However, due to the complexity of the real-world food system, there are a small number of expected outliers, notably Japan and the Republic of South Korea, both of which score in the range 0.4 to 0.6, much lower than expected given their level of industrialisation. This can be explained by a combination of two factors: a slightly lower post-production food waste of around 15% (Liu et al., 2016) and typically a

lower daily calorific intake than other similarly industrialised nations; a result of cultural and dietary trends (Tsugane and Sawada, 2014).

The parameter is used in the model processes to inform processes relating to agricultural efficiency, including food energy losses at three stages: processing, distribution, and post-production losses. The ratios of livestock feed energy obtained from forage and non-forage are also derived using this parameter, along with the portion of food waste that is used as livestock feed. Minimum and maximum values are chosen for each, representing either the totally subsistence or total industrialised case, and the metric is used to scale the value for a country between the two. The equation for a factor $\mu$ is:

$$\mu_a(n) = \mu_{\text{sub}} + X_a(n)(\mu_{\text{ind}} - \mu_{\text{sub}}) \tag{6}$$

where $X$ is the value of the food system efficiency parameter for the country $a$ in given year $n$ and $\mu_{\text{sub}}$ and $\mu_{\text{ind}}$ are the subsistence or industrialisation boundaries of the factor respectively. The upper and lower boundaries for each of these parameters can be modified as a means of scenario adjustment. The behaviour of the boundaries as the model progresses can also be modified; they can be fixed at the initial values, or an overall efficiency increase can be prescribed, in which case the

185 limits will also change over time.

### 3.2.2 Inefficiency in the food system

In C-LLAMA, losses in the food system are grouped in four ways: losses at the harvest stage, losses in the processing stage, distribution losses and post-production losses.

Losses at the harvest stage occur before any processing or distribution and are either non-recoverable or recoverable. Causes
of non-recoverable losses include insect and animal pests, weeds, and disease. Developing regions see greater losses during production than developed regions due to the availability of disease and pest prevention measures (Oerke and Dehne, 2004; Savary et al., 2012). Losses due to these factors are accounted for in crop-yield data so no loss factor is applied at this stage.

The methodology for handling recoverable harvest losses: 'harvest residues', is more complicated since these are crop dependant. Not all harvested material is edible for humans, for example the husks and casings or `chaff' produced when
harvesting grains. The formalisation of this concept is the harvest index, defined as the ratio of the mass of useful product to the mass of above ground biomass (Singh and Stoskopf, 1971). Despite being an inefficiency in the food system, many waste products produced at the harvest stage can be used for other purposes to reduce this inefficiency. Chaff for example, while inedible to humans, is suitable feed for most livestock. Harvest residue indices and harvest residue recovery rates are used to inform a ratio of produced residue to recovered residue (Krausmann et al., 2008; Wirsenius et al., 2010). Tables of harvest
residue indices and recovery rates can be found in appendix B.

Processing losses occur as the raw crops are processed to a form suitable for their intended purposes, for example the removal of kernels from olives. Some of these losses are potentially recoverable for use as animal feed, bioenergy feedstock or in other industries (Van Dyk et al, 2013). Fodder crops generally incur less loss than crops destined for human consumption at the processing stage as they require little to no processing (Gustavsson and Cederberg, 2011; Kitinoja, 2013).

Distribution losses are incurred through transportation or storage. This stage is a major contributor to food system inefficiency in developing countries; due to poor road infrastructure, pests and lack of suitable refrigeration or other storage, losses at this stage can be as high as 50% and as low as 5% in developing and developed areas respectively (Lipinski et al., 2013; Parfitt et al., 2010).

Post-production food waste refers to food lost at the consumer level, including food thrown away after purchase in the home, or in commercial environments such as restaurants. Unlike most other food system loss factors, the heaviest post-production losses are seen in the developed world (Parfitt et al., 2010; Stancu et al, 2016). Since post-production waste is inherently included in food supply data, the post-production factor shown in table 1 is used only to estimate the amount of post-production waste potentially available for use as livestock feed.

| Loss factor | Industrialised (1.0) | Subsistence (0.0) |
| --- | --- | --- |
| Processing | 6% | 10% |
| Distribution | 5% | 50% |
| Post-production | 30% | 5% |
| Post-production waste to feed | 5% | 40% |
| Other waste to feed | 40% | 15% |

**Table 1. Boundary values for factors informed by the food system efficiency parameter.**

### 3.3 Food production

#### 3.3.1 Production

Following the application of the loss factors determined in the food system efficiency section to the food supply projections described in section 3.1, each country is left with a food energy requirement for each food commodity, $r$, calculated using the following equation:

$$r_{j,a}(n) = \frac{E_{j,a}(n)}{\prod_{l \in L}(1 - \mu_{l,a}(n))} \tag{7}$$

where $r$ is the energy demand from a country $a$ for commodity $j$, $\mu$ is a loss factor and $L$ is the set of processing and distribution losses. $E$ is the calorific contribution to the countries food supply from commodity $j$, described in section 3.1. The food energy lost due to efficiency loss factors is retained for potential re-use as livestock feed. Food demand is then summed globally for each key commodity or commodity group is, so the global production requirement $R$ for the commodity $j$ is

$$R_j(n) = \sum_{a \in A} r_{j,a}(n), \tag{8}$$

where $r$ is the food energy demand for commodity $j$ from a country $a$, and $A$ is the set of all countries.

C-LLAMA does not have a formal representation of trade, instead trade is implicit in the allocation of food production; global proportions of production for each crop commodity are calculated using the most recent five years of production data then allocated accordingly. For example, the USA was responsible for 42% of global wheat production between 2012 and 2017,

thus 42% of all wheat production in C-LLAMA is allocated to the USA. To account for the significant industrial use of primary crops in Brazil and the USA, the historical production value is reduced by a factor to provide an estimate for only food use of those crops. These factors are 0.34 and 0.289 for sugar cane in Brazil and corn in the USA (Bordonal et al., 2018; De Miranda and Fonseca, 2019; Mohanty and Swain, 2019). Following this process, each nation is left with a *production allocation* for each key commodity and commodity group, the equation for which is

$$q_{j,a}(n) = \frac{M_{j,a}}{\sum_{a \in A} M_{j,a}} * R_j(n) \tag{9}$$

where $q$ is the allocated production energy of commodity $j$ in the country $a$, $M$ is the mean of the most recent five years (2012 to 2017) of historical production mass of commodity $j$ in country $a$ and $A$ is the set of all countries.

### 3.3.2 Crop yield

A large proportion of yield variation can be explained by climate variability, with the remainder being a result of farming practices and industrialisation (Mueller et al., 2012; Ray et al., 2015). C-LLAMA takes largely the same approach as FALAFEL; historical yields for each crop and group are projected linearly to 2050, but this is done for each country. Yield has the potential for large transient variation on a year by year basis, often a result of climate events, pests or management (Frieler et al., 2017; Ray et al., 2015). Consequently, there is the possibility of yields increasing at an unrealistically high rate through this kind of projection. To address this, in C-LLAMA yields are capped at the historical maximum value for a region, preventing any region from exceeding an observed value whilst allowing each country within a region to catch up to a localised observed maximum. Linear projections with a p-value greater than 0.05 (this threshold can be changed) or a decreasing yield are discarded. In either of these cases, the mean yield from the previous ten years of data is used instead.

For all key crops the raw yield data, in tonnes per hectare per year, was used to make the projection. In the case of grouped crops, the groups yield was calculated by taking mean of all crops contained in the group, weighted by national production mass. The group 'sugar crops' consists almost entirely of sugar beet since sugar cane is represented as an individual crop. For palm oil, vegetable oils and other oil crops, an effective oil yield was calculated for each using their respective oil factors which can be found in the FAOSTAT database (Food and Agriculture Organization of the United Nations, 1997).

### 3.4 Livestock

Animal product demand is one of the highest contributors to agricultural land demand and greenhouse gas emissions globally, with estimated emissions between 5.6 and 7.5 Gt $CO_2$ $yr^{-1}$ equivalent between 1995 and 2005; as such livestock are a crucial component of the C-LLAMA model (Herrero et al., 2016; Pikaar et al., 2018; Van Zanten et al., 2018). As with vegetal food commodities, livestock commodities are partially grouped, with major commodities: bovine meat, pig meat, mutton/goat meat and poultry meat remaining separate. The remaining meat products contribute comparably little to the global demand for animal products and are grouped into an 'other meat' category. Eggs, dairy and fish are each in their own groups. For each country, an animal commodity demand is produced per year in the diet and food supply section of the model. As is well

established, livestock are inherently less resource efficient than vegetal products as a means of providing calories for human consumption. The feed consumed by livestock does not go directly to become fresh animal product, instead much of it supports the survival of the animal. This is commonly quantified as a feed efficiency (FE) or livestock conversion efficiency (LCE, the inverse of feed efficiency), expressed as the quantity of fresh animal product to feed energy mass or equivalent energy. This number varies drastically between animal product types: bovine meat has an energy FE of approximately 3%, whereas poultry meat is much higher at 21% (Shepon et al., 2016). Note that these FEs are produced from data acquired in the USA. Currently the values used in C-LLAMA are taken from FALAFEL; a cohesive energy-equivalent FE dataset was not found at a regional or country level. FEs certainly do vary regionally, largely due to the different role of livestock in different food systems. A cow in a subsistence agriculture environment is more likely to be allowed to live to substantial age, providing dairy and driving machinery. This contrasts with a cow in industrialised agriculture, where it might be reared solely for meat and slaughtered in early adulthood (Wirsenius et al., 2010). A proportion of livestock feed demand is met through forage ($\mu_{\text{forage}}$) and the remainder is met through feed and residues ($\mu_{\text{non-forage}}$, equivalent to $1 - \mu_{\text{forage}}$), calculated using the food system efficiency parameter to assign a value between the subsistence case and the industrialised case, using the same method as in Eq. (6). The quantity of feed demand energy from non-forage $D$ for animal product $j$ in country $a$ and year $n$ is

$$D_{j,a}(n) = Q_{j,a}(n) * \mu_{\text{non-forage } j,a}(n) * \frac{1}{\text{FE}_j},  \tag{10}$$

where $\text{FE}_j$ is the livestock dependant feed efficiency and $Q$ is the production allocation. The extreme cases for each animal product are centred around the FALAFEL numbers, with the developing limit being 20% lower and the developed limit being 20% higher. The proportion also varies dependant on the animal product, for example chickens and pigs typically obtain a higher proportion of their food energy from feed than ruminants (Tufarelli et al., 2018). An individual animal will likely be fed through a combination of forage and feed, but for the purpose of the model the assumption is made that the land footprint of non-foraging animals comes only from the land required for fodder crops. The portion of livestock feed demand met through forage is therefore $\frac{1}{\text{FE}_j} * Q$ minus $D$ for each animal product $j$. This approach is coarse compared with modelling livestock as entities with individual mixed feed demands, however the feed energy requirements are comparable.

### 3.4.1 Waste and residues as feed

In some situations, livestock can utilise waste from the agricultural system, processing losses, post-production food waste and harvest residues. For each livestock commodity a potential feed ratio for each of these waste streams is estimated: the maximum proportion of each waste type that could contribute to the livestock diet ($z$). These ratios can be found in appendix C. Waste produced by processing, distribution and post-production are calculated at the country of consumption, while harvest residues are calculated at the crop production stage. Post-production waste is assumed to only be available to animals in the area in which it was produced and is informed by a post-production waste to feed factor ($\mu_{\text{post}}$), scaled by the food system efficiency parameter using Eq. (6) between 40% and 5% for the subsistence and industrialised cases respectively. Note that in the case

of post-production waste the subsistence extreme is 'more efficient' than the industrialised case. The remaining total available waste energy is multiplied by an 'other waste to feed factor' ($\mu_\text{other}$), again informed by the food system efficiency parameter using Eq. (6), with the subsistence and industrialised limits being 15% and 40% respectively. Other waste is that of harvest residues and processing waste, but not distribution waste since this is 'lost' or spoiled. These numbers are taken from the low and high efficiency scenarios in FALAFEL. Waste energy is 'fed' to livestock, up to the potential feed ratio limit, allocated by the potential feed ratios ($z$). The energy used is then subtracted from the livestock feed energy demand, the remainder of which is accounted for with fodder crops. The remaining feed energy demand to be met through fodder crops ($D'$) is

$$D'_{j,a}(n) = D_{j,a}(n) * \left[1 - \sum_{\omega \in \Omega} z_{j,\omega}\right] + \sum_{\omega \in \Omega}\left[S\left(D_{j,a}(n) * z_{j,\omega} - \left[w_\omega(n) * \mu_\omega * \frac{z_{j,\omega}}{\sum_{c \in C}(z_{c,\omega})}\right]\right)\right] \tag{11}$$

$$S(x) = \begin{cases} x & x > 0 \\ 0 & x \leq 0 \end{cases} \tag{12}$$

where $D$ is the total feed energy demand, $z$ is the maximum portion of feed energy that livestock $j$ can obtain from waste stream $\omega$, $w$ is the available waste energy and $\mu$ is the waste to feed factor. $C$ is the set of all livestock commodities and $\Omega$ is the set of all waste streams: post-production, processing, and harvest residues. $\mu$ is $\mu_\text{post}$ for post-production waste and $\mu_\text{other}$ for all other waste streams.

**3.4.2 Fodder**

Following the reduction of livestock feed demand through waste to feed and foraging, the remaining feed energy demand is met with fodder crops. The historical fodder mix, the ratio of each crop making up fodder in a country, is calculated using the most recent five years of 'feed' energy data in the FAOSTAT food balance sheets. The cereals contributing the most to the fodder mix globally are maize, wheat, sorghum, barley and rice. In addition, soybeans, potatoes, cassava, pulses and fruits also contribute in the top ten. Each of these products are represented individually while all other products used as feed are grouped as 'other feed'. Around 8% of the total feed mass each year comes from non-crop products. The majority of this 8% is milk and the remainder is largely comprised of aquatic products such as fishmeal and aquatic plants, often added to livestock feed to supplement nutrition (Holman and Malau-Aduli, 2013; Oliveira Vieira et al., 2015). These products are removed from the fodder mix, as these products require minimal additional land. The remaining livestock feed demand is split according to the derived fodder mix, so the contribution to the total fodder requirement ($r$) in country $a$ from fodder product $k$ is

$$r_{k,a}(n) = \frac{f_{k,a}}{\sum_{s \in S} f_{s,a}} * \left(1 - \frac{f_{\text{milk},a} + f_{\text{aq},a}}{\sum_{s \in S} f_{s,a}}\right) * \sum_{c \in C}(D'_{c,a}), \tag{13}$$

where $f$ is the five year mean of feed data for fodder product $k$ from the FAOSTAT food balance sheets, $f_\text{milk}$ and $f_\text{aq}$ are the feed data for milk products and aquatic products respectively, $S$ is the set of all fodder products. $D'$ is the fodder demand for livestock commodity $c$, $C$ is the set of all livestock commodities. The global production requirement for fodder product $k$ is then

$$R_k(n) = \sum_{a \in A} r_{k,a}(n). \tag{14}$$

In the same way as crop production for food, the fodder crop production demand is allocated based on historical production of the fodder products. The production allocation ($q$) for fodder product $k$ for country $a$ is

$$q_{k,a}(n) = \frac{M_{k,a}}{\sum_{a \in A} M_{k,a}} * R_k(n), \tag{15}$$

where $M$ is the five year mean production mass for fodder product $k$ and $A$ is the set of all countries. In the case where the product has been considered as a food commodity and thus a yield and production allocation has already been calculated, the additional production allocation for fodder is simply added to the nations existing production quota of the commodity for food. In some cases, it is necessary to perform a yield projection in the same manner as described in section 3.3. Following this stage, each country has a production quota for each year for each commodity, used for food, animal feed, or both, along with a corresponding yield trajectory.

### 3.5 Land use

### 3.5.1 Crop land use

A simple division of yearly crop production allocations by national crop yield projections produces a yearly land demand trajectory for each crop within a given country. Since the model objective is to explore sensitivities rather than absolute land-use values, land-use is projected from the most recent value in the FAOSTAT data: a calibration factor is used to align the 2017 value of the projected values with the 2017 historical value, for each crop. In the case that total land demand for crops is less than the previous year, the land difference between the years is put into a 'freed land' class. In FALAFEL this land is then used for either afforestation or energy crops, while C-LLAMA does not currently process this further. In reality land use change is multidimensional; the abandonment of agricultural land varies greatly between areas and industrialisation levels, influenced by climate, land productivity, tradition and governance (Lambin et al., 2003; Lambin and Meyfroidt, 2011). C-LLAMA currently does not consider non-agricultural land use. Further development to include more complex handling of land-use is intended.

### 3.5.2 Livestock land use

As mentioned in section 3.4, the land requirements for livestock (in addition to fodder crop production) in C-LLAMA come entirely from their pasture area; the implication being that all fodder fed animals are under roof, while their foraging counterparts graze pasture. This is generally not the case for foraging pigs and chickens, so a pasture factor ($\rho$) of 0.1 is applied to reduce their land footprint from that of cows and sheep (Tufarelli et al., 2018).

The land used for livestock pasture is calculated using an *effective pasture yield.* First, the historical energy obtained from pasture by livestock was estimated using a similar process to the method adopted in Haberl (2007); for each country, available feed is subtracted from a livestock feed demand, calculated using historical production energy and feed conversion ratios between 1961 and 2017. This leaves animal food acquired through forage. Dividing this quantity by land-area used for pasture in a given year results in the historical effective pasture yield – animal product energy produced per hectare of pasture. The

land-area data used is taken from the FAOSTAT database (Food and Agriculture Organisation of the United Nations, 2021). The historical effective pasture yield ($Y$) for animal products in country $a$ is

$$Y_a = \frac{1}{L_{\text{pasture},a}} * \left( \sum_{j \in J} [M_{j,a} * \text{FE}_j * \rho_j] - \sum_{k \in K} f_{k,a} \right), \tag{16}$$

where $L_{\text{pasture}}$ is the country's pasture land area, $M$ is the production mass of an animal product $j$, $\text{FE}_j$ is the feed conversion ratio for the animal product and $J$ is the set of animal products. $f$ is the quantity of available feed product $k$ and $K$ is the set of all feed products. The historical trajectory is linearly projected to 2050; the pasture yield and pasture production mass demand together give a projected pasture land requirement for each livestock commodity. Since there is no historical data to calibrate the yield value to, the yield value is scaled such that the projected 2017 pasture land-use matches the 2017 historical pasture land-use. The value is calibrated to the anchor scenario described in section 5, rather than being scenario specific, to address counter-intuitive model behaviour, discussed in appendix F. Because this can result is minor discontinuity when running non-anchor scenarios, the projected land-use is then calibrated to the historical land-use too. This method is coarse but offers a catch-all method of translating a production demand into land-area for every country in C-LLAMA.

## 4 Model output

C-LLAMA produces a land-use trajectory from 2013 to 2050 for each food commodity and commodity group within a country, output as a comma separated variable file. Animal product land-use is aggregated as pasture, explained in section 3.4. All crops have individual land-use trajectories. An output with crops aggregated into either crops or specifically fodder crops is also produced. Data from intermediate stages of the model such as food supply, production, and crop yield projections is retained upon completion of the model run. However, given that calibration of the model occurs at the final stage rather than at every intermediate stage, these trajectories should be viewed with this in mind. Food supply and crop yield projections are both direct projections of historic data and so are exempt from this. For the sake of model run time, intermediate outputs are stored in a serialised format using the 'pickle' library, part of the Python standard library (Van Rossum and Drake Jr, 1995).

### 4.1 Anchor Scenario

C-LLAMA is based around an anchor scenario, in which all parameters take default values based on literature and projections from historical data are made to 2050. This scenario aims to be as close an approximation to the real world as possible in the framework of the model, with targets for efficiency and industrialisation being set at middle of road values. Table D1 in shows key parameters and their values in the anchor scenario. Regionally aggregated land-use types in the anchor scenario can be found in appendix E.

Figure 2 shows agricultural land-use at the continental level for historical FAOSTAT data and in the C-LLAMA anchor scenario. All continents aside from Oceania see an increase in land-use for both crop and animal production, with the rate of increase slightly decreasing toward 2050, particularly in Africa. The greatest rate of increase occurs in Asia and the least in

Africa and Europe. In all cases, the rate of increase for pasture is greater than that of cropland, with cropland for fodder crops lying in between. The direction of the projected land-use aligns with that of the historical data in the Americas, Africa, Oceania, and Asia. However, in Europe a slight reversal of the direction of change occurs, a result of the significant historical production of beef and dairy production in Russia; Russia produced 4% of the World's bovine meat in 2013, hence is allocated a significant portion of beef production in the model processes and resultant pasture area increase

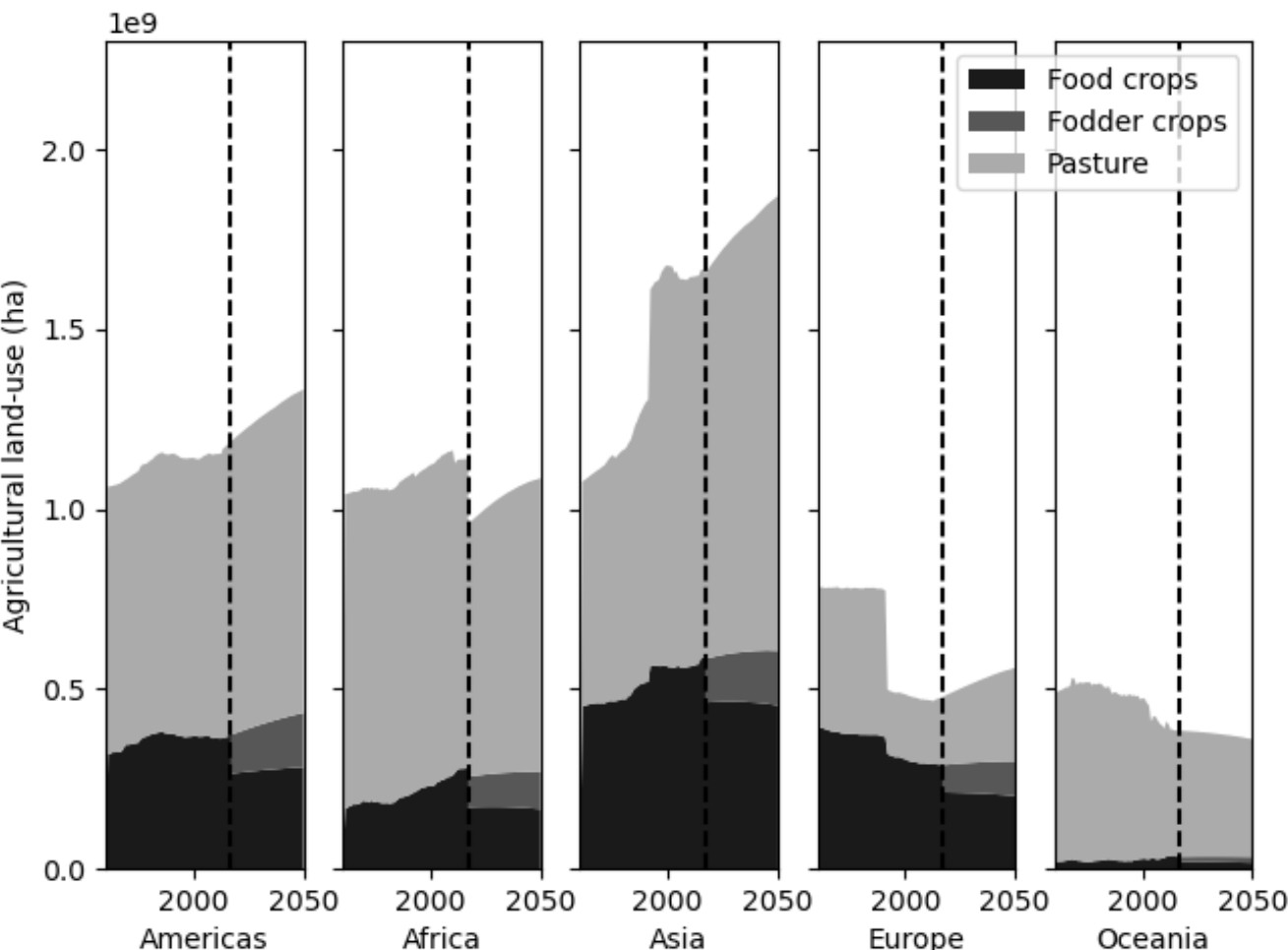

**Figure 2. Agricultural land-use in FAOSTAT historical data and C-LLAMA anchor scenario projection for five continental regions. The transition from historical to modelled data is denoted by the dotted black line. Discontinuity at the dotted line is due to the countries not included in C-LLAMA for various reasons described in section 3. 99.7% of this discrepancy is the result of unavailable food balance data for Libya, Somalia, Sudan, the DRC, and Papua New Guinea. Also note the sudden increase of land-use in Asia and corresponding decrease in Europe in the early nineties,**

**the result of the dissolution of the Soviet Union. As the FAOSTAT land-use does not contain disaggregated crop-data for fodder and food, food crops also include fodder crops in the historical data.**

Figure 3 shows the projection of mean diet at the continental level in the C-LLAMA anchor scenario. All continents undergo an increase in total calorific intake toward 2050. With the magnitude of change being similar at around 400 kcal for every

continent with the exception of Europe, which sees a lesser increase of approximately 200 kcal by 2050. The proportional increase varies however, with the greatest proportional increase occurring in Africa. The consumption of non-egg and dairy animal products increases in across all continents, although only slightly in Africa. The consumption of cereals decreases slightly in Asia and Europe, but increases slightly elsewhere, with the strongest increase in Africa. The demand for oil crops sees similarly proportional increases in every continent, with Europe and Oceania consuming more.


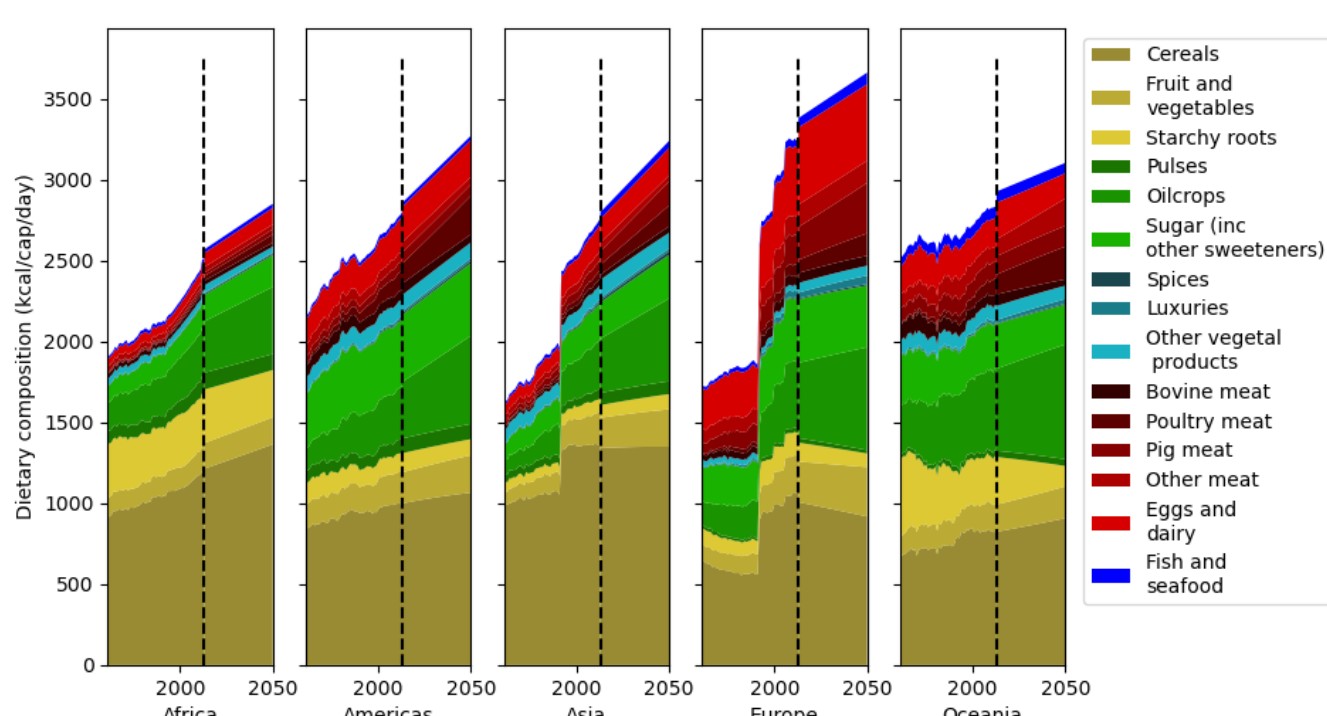

**Figure 3. Calorific mean diet composition at the continent level in historical FAOSTAT data and the C-LLAMA anchor scenario. Some food commodities are grouped for clarity and the order of appearance from the origin for the groups**

**aligns with the legend.**

### 4.1.1 Comparison with FALAFEL

The globally summed land-use output of the C-LLAMA anchor scenario can be compared with the land-use trajectory of an analogous business as usual scenario produced in FALAFEL. In the same way as C-LLAMA, the FALAFEL model allows prescribed increases in efficiency – for example a forced reduction in animal product consumption. To produce the business
as usual scenario in FALAFEL, linear projections are made where they are available and all prescribed efficiency changes are turned off.  For comparison, the land-use data from both models is grouped into pasture, food crops (for human consumption), and fodder crops. The resulting land-use for both modelled scenarios is shown in Fig. 4. The trajectory of both the FALAFEL scenario and the C-LLAMA anchor scenario reach just over 5 Gha by around 2050, with C-LLAMA reaching approximately 5.2 Gha, an increase of approximately 450 Mha. The difference in starting food crop area is slightly higher in C-LLAMA, and
a small amount of additional growth occurs by 2050 in C-LLAMA. C-LLAMA starts with a lesser area of fodder crops but sees less proportional growth by 2050 than in FALAFEL. Both models see an increase of approximately 90 Mha in total cropland by 2050. The largest difference lies in pasture, with C-LLAMA starting at just over 3 Gha and FALAFEL starting at around 2.6 Gha. Both models have a very similar pasture area in 2050 around 3.4 Gha. The method used to estimate pasture area in FALAFEL is completely different to that of C-LLAMA, using estimates of land-productivity and energy uptake by
livestock, rather than calculating an empirical pasture-yield.

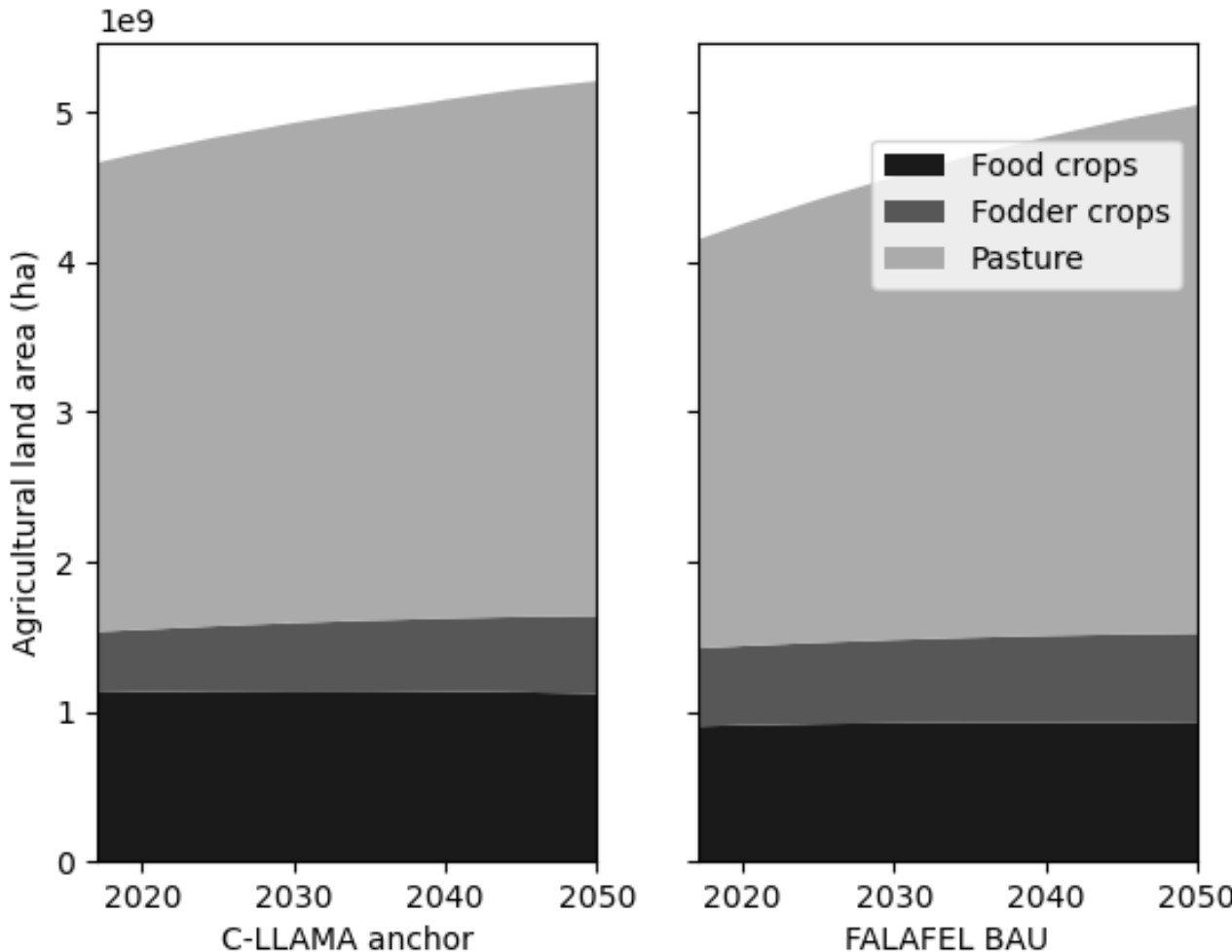

**Figure 4. Aggregated global land-use for food production in the C-LLAMA anchor scenario and a 'business as usual' (BAU) FALAFEL scenario. FALAFEL accounts for the production of some non-food crops, however they are excluded**

**for this comparative figure.**

### 4.2 Sensitivity

Four key projections are made throughout the course of the model for each country. Diet and crop yields are projected directly from the historical data, whereas the food system efficiency parameter and effective pasture yield are internal values calculated from historical data, which are then projected. To explore the sensitivity of the final land-use output of C-LLAMA to these

four projections, each was fixed at the mean value of their most recent five years and the land-use by 2050 compared with the anchor scenario. The results of this are shown in Fig. 5.

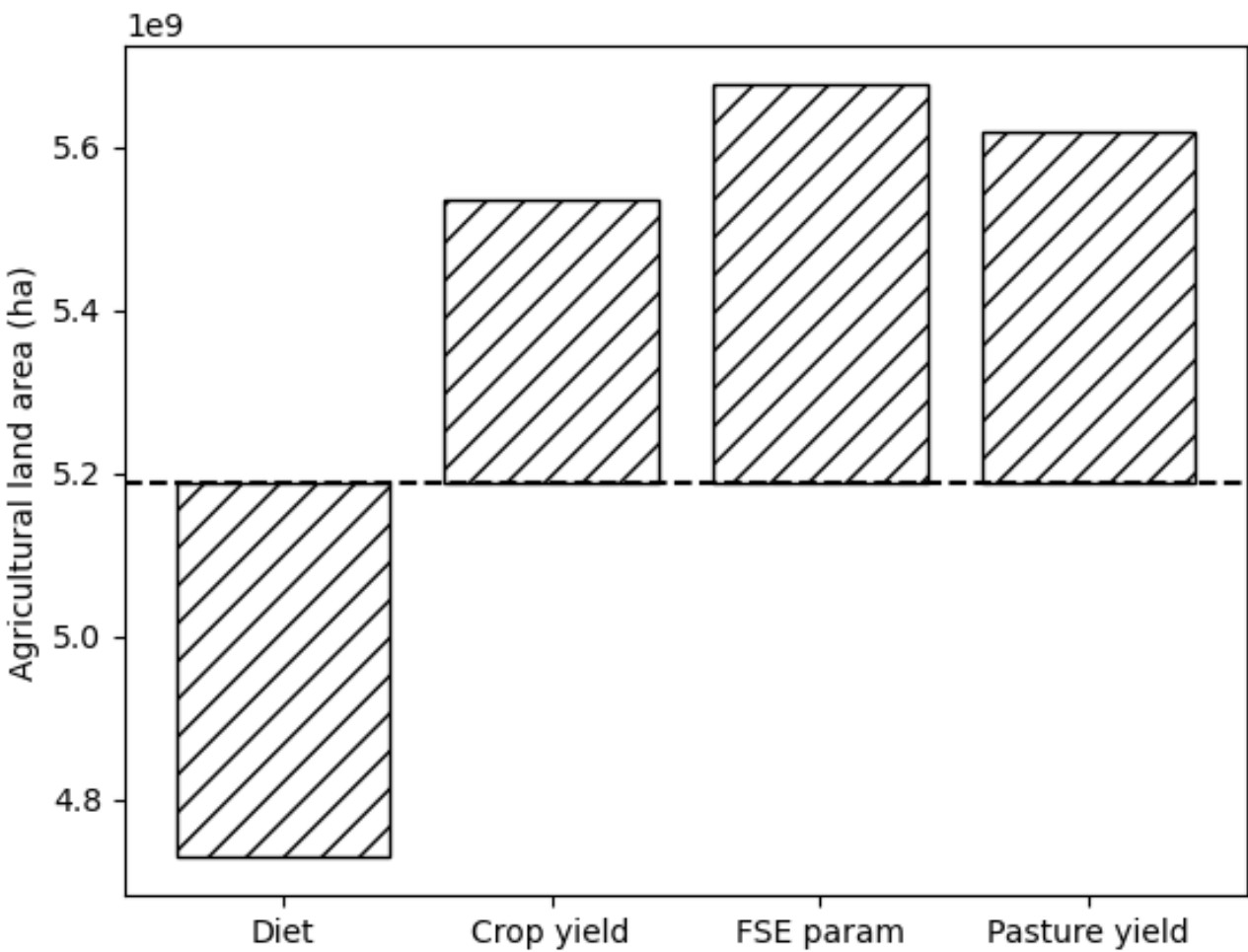

**Figure 5. Difference in 2050 global agricultural land-use between the anchor scenario (dotted line) and when disallowing the progression of projections in the model by using the 5 year mean of historical values for each.**

The impacts of each of these projections are within an order of magnitude of each other. Halting the projection of crop yields results in an increased agricultural land-use of approximately 300 Mha from the anchor scenario. This is consistent with the current trend of increasing crop yields in most areas of the world: a result of improving access to irrigation, agrochemicals and machinery (Iizumi et al., 2017; Ray et al., 2012). Suspending the projection of the food system efficiency parameter has the greatest impact on the total land-use with an increase of approximately 500 Mha. Suspending the food system efficiency parameter locks many countries in a state of lower efficiency, unable to meet the increasing food demand from the growing population. Halting changes in pasture yield leads to an increase in land-use of around 450 Mha. While the 'effective pasture yield' is not a real-world quantity, it aims to capture a wide range of factors that govern the output of grazed land. This quantity is increasing in the majority of countries, the result of livestock intensification by transfer to more intensive pasture or a covered system (Davis and D'Odorico, 2015; Thornton, 2010). Stopping the projection of dietary trends reduces the final land-use by

approximately 450 Mha. Current global dietary trends are toward increased animal product consumption in developing countries and stagnation of animal product consumption in developed nations (Tilman and Clark, 2014; Van Zanten et al., 2018). This combined with an increase in total calorie intake in the majority of countries explains the decrease in land-use when suspending the projection of diet.

Loss factors in C-LLAMA are dynamic, governed by the food system efficiency parameter. To explore the sensitivity of the model to loss factors every country was fixed at the lower and upper boundary values, equivalent to scoring every country at 0.0 or 1.0 respectively on the food system efficiency parameter. Figure 6 shows the results of this analysis, along with a fully vegetarian (by 2050) diet scenario. Scores of 1.0 leads to a land-use increase of approximately 700 Mha by 2050, and a global score of 0.0 leads to an almost identical increase of just over 700 Mha by 2050. Scores of 1.0 and 0.0 both precipitate very

high loss ratios from the start of the model of around 30% in post-production and production respectively. The present efficiency scenario is achieved by setting the food system efficiency parameter at it's present values, identical to the 'FSE param' scenario in Fig. 8. The fully vegetarian diet scenario sees a drastic land-use decrease of approximately 1.8 Gha by the year 2050, which is consistent with the previously discussed effective land-use inefficiency of animal products as food when compared to vegetal products.


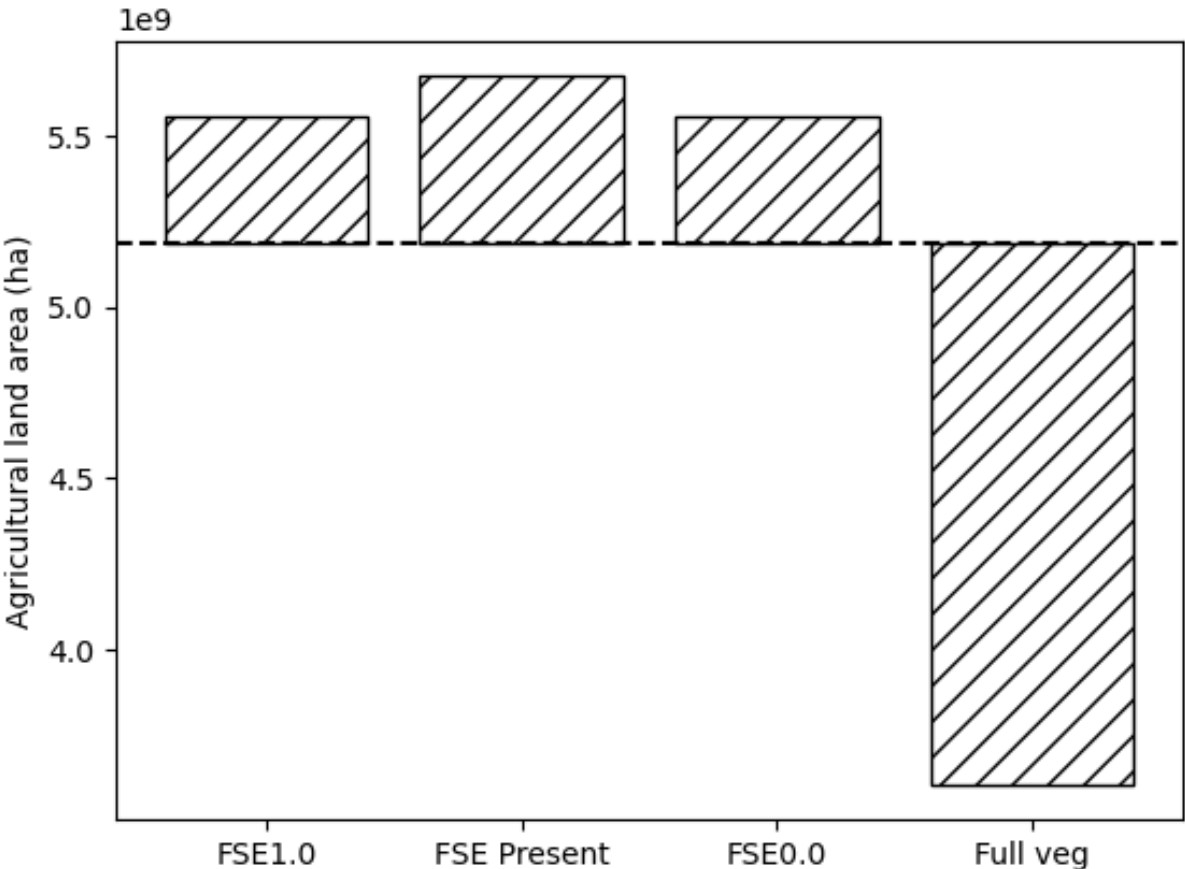

**Figure 6. Change in 2050 agricultural land-use between the anchor scenario (dotted line), maximum, present, minimum efficiency, and full-vegetarian diet scenarios. Maximum and minimum efficiency scenarios are produced by setting the food system efficiency parameter to 1.0 and 0.0 respectively for all countries. The full-vegetarian diet scenario tends toward a 100% plant-based diet globally by the year 2050.**

**5 Discussion**

Estimates of historical agricultural land cover, cropland harvests, and land-use change are plentiful (Erb et al., 2017). There are a wide range of approaches from book-keeping to satellite imaging, the majority of which are available at high spatial resolutions (Fritz et al., 2015; Hurtt et al., 2011; Winkler et al., 2021). These datasets are used as starting points for other modelling approaches such as IAMs or vegetation models but cannot be used to directly make projections of land-use. From these starting points, a great number of model and scenario drivers impact the land-use trajectories of IAMs, including economy, energy demand, commodity pricing and policy. IAMs are excellent tools for making holistic projections about a wide range of factors in given scenarios, but the land-use component is difficult to extract. The purpose of C-LLAMA is to

explore the sensitivity of agricultural land-use to various drivers within the food system, not to make explicit predictions about land-use within specific countries.

The C-LLAMA anchor scenario projects cropland and pasture land-uses of approximately 1.64 Gha and 3.57 Gha respectively by 2050. The projected cropland value is within the range of projected values from IAM scenarios in the comparable SSP2 and broader AR5 databases, shown in Table 2, and well within estimates of cropland availability (Eitelberg et al., 2015). However, the projected pasture value is slightly outside the range of other SSP2 scenarios, albeit only 70 Mha greater than the marker scenario. The majority of agricultural land expansion in SSP2 scenarios occurs in Africa and Latin America (Popp et al., 2017). In C-LLAMA there are pasture expansions in these regions, along with expansion occurring in North America and Asia, due to the very limited trade mechanics of C-LLAMA. Note that the scenarios in these databases are based around key assumptions and pathways in the social and economic sectors, whereas the only prescribed trajectory within C-LLAMA is of population. As previously discussed, the intention of C-LLAMA is not to predict land-use futures, so this behaviour in these regions does not diminish the efficacy of the model as a means to explore sensitivities to drivers.

| | Cropland 2050 (Gha) | | | Pasture 2050 (Gha) | | |
|---|---|---|---|---|---|---|
| | Marker | Min | Max | Marker | Min | Max |
| C-LLAMA | 1.64 | | | 3.57 | | |
| SSP2 scenarios | 1.76 | 1.60 | 2.18 | 3.53 | 2.47 | 3.53 |
| AR5 database | 2.10 | 1.27 | 3.33 | 3.83 | 2.67 | 4.72 |

**Table 2. Global cropland and pasture land cover in the year 2050 in C-LLAMA, SSP2 scenarios, and the AR5 scenario database. The mean of all AR5 scenarios is used for the AR5 database marker value.**

A fully vegetarian scenario in C-LLAMA sees a significant decrease in agricultural land-use of 1.8 Gha (a reduction of approximately 34%), in-line with the literature (Röös et al., 2017; Swain et al., 2018; Weindl et al., 2017; Van Zanten et al., 2018). The nutritional implications of such a diet were not considered in this scenario; which is likely to be a significant hurdle in the transition to sustainable diets (Duro et al., 2020; The Eat-Lancet Commission, 2019). With the ability to prescribe trajectories for diet at a country level, C-LLAMA is well placed to explore such questions. Nutritional information could also be built into C-LLAMA for each commodity.

The strength of C-LLAMA lies in its simplicity: it can be easily modified, adapted, and improved. However, there are limitations to the approach and two key areas for improvement have been identified. One area with scope for improvement is in the allocation of crop and livestock production described in section 4.3. The current method uses a snapshot of current production to distribute the projected production of a crop; this approach works for earlier projected years since interannual changes to trade are relatively slow, being on similar timescales to changes in demand. However, long term changes to global trade are not captured, specifically those likely to arise from improved access to wealth and subsequent demand for luxury and animal products in developing countries. Improvements might include trade matrices for each food commodity, or a forward

projection of the commodity production allocation, which would allow semi-dynamic trade representation without the need for any agent based or economically driven modelling. The other area with great potential for improvement is the representation of livestock and, more broadly, land-use within the model. The current method for estimating land-use for crops and livestock is effective for exploring questions surrounding global-scale changes and scenario options. However, a land class system with productivity, land-use transitions, and associated carbon exchange would facilitate a more nuanced exploration of the drivers of land-use and their consequences, particularly in the case of livestock, forests, and grasslands.

Including the DRC, Libya, Sudan, Somalia, and Papua New Guinea would be beneficial as together they account for a significant portion of the global land area (approximately 3%). Papua New Guinea and the DRC have humid, equatorial climates with highly productive land; excellent conditions for agricultural productivity (Kottek et al.; 2006). While not included in the food balance data, they are present in other FAO data, so it may be possible to construct an approximate food balance dataset from their available FAO data and regional averages. Another approach would be to construct food balances using other data sources, however this approach would contravene the internal consistency of C-LLAMA.

C-LLAMA takes a simple approach to modelling the drivers of land availability, offering transparency and adaptability where more complex modelling approaches do not. Of the many drivers of future land-availability, the simplicity and traceability of the model make it well placed to explore the impacts of broad scale drivers such as changes in livestock production systems, crop yields, dietary trends and food system efficiency on the future of land available for food agriculture, bioenergy and afforestation from a bottom-up perspective. For example, scenarios with prescribed increases to crop yields, consumption of specific commodities, calorie intake, or wasted food could be constructed. The structure of C-LLAMA also facilitates that these changes can be applied at regional or country levels. The model aims to be easily accessible to use and modify, using only open source data and software.

**Appendix A**

| Model Section | Description of processes within the section | Relevant modules |
|---|---|---|
| Diet and food supply | Projections of the contribution of each food commodity toward the national diet. Projection of national calorie supply per capita. Calculation of a global demand for each food commodity. | food_demand_and_waste_production diet_makeup |
| Food-system efficiency | Projection of losses and efficiencies that are used at various stages of the model. An food | industrialisation_metric industrialisation_metric_calculations food_waste_gen |

| | | system efficiency parameter is developed to inform these values. | harvest_residues |
|---|---|---|---|
| | Crop production | Losses are used to calculate total production requirement for each food commodity, a portion of which is then allocated to each country. Crop yields are projected for each food and fodder crop in the model. | crop_yield_and_production_hist<br>crop_yield_and_production_params<br>crop_yield_projects<br>crop_and_livestock_production<br>crop_production_ratios |
| | Livestock | The global production requirement for livestock is calculated and allocated to each country. Livestock consume a mix of feed and foraged food, the proportion coming from each varies by livestock type and country. | livestock_feed_demand<br>fodder_crops |
| | Land use | Production requirement energy is converted to mass and combined with yield to produce a land area requirement for both food and fodder crops. An 'effective livestock yield' is developed and used to calculate pasture land requirements. | crop_land_calculations<br>pasture_land_calculations<br>land_use_calculation |

**Table A1. Five main sections within C-LLAMA, each comprised of a handful of model-process modules. There are sixteen model-process modules in total.** There is some overlap between model-processes; the sections and model-process modules listed here are not necessarily in the order that they appear in C-LLAMA, some sections are re-visited at later stages of the model. The first section of the model produces a food supply at a national level, disaggregated into calories and commodities.

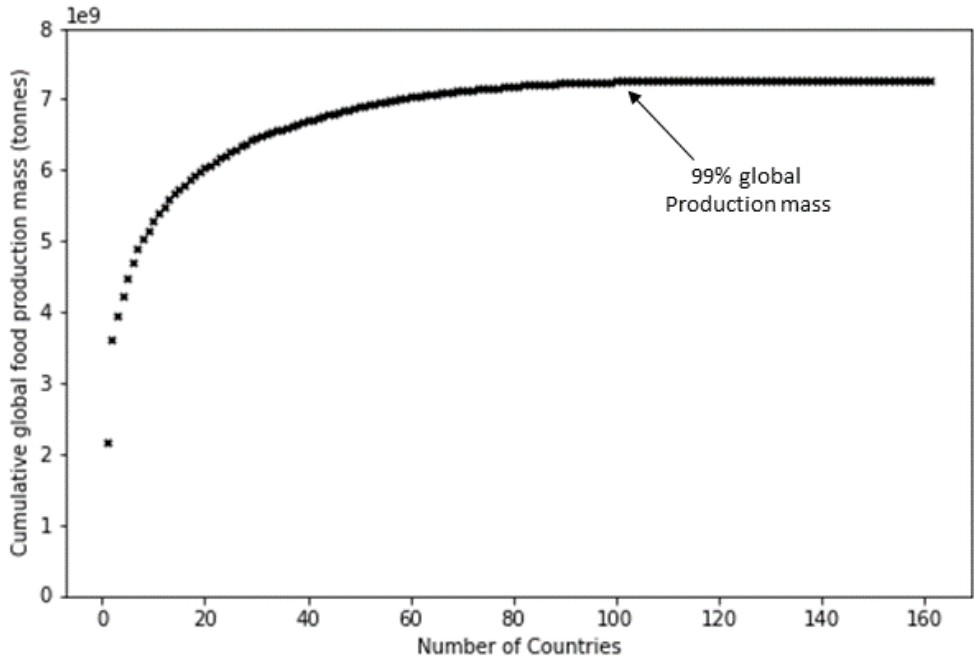

**Figure A1. Cumulative food production mass for the year 2017 of all current countries in the FAOSTAT database,**
**dissolved states are not included.**

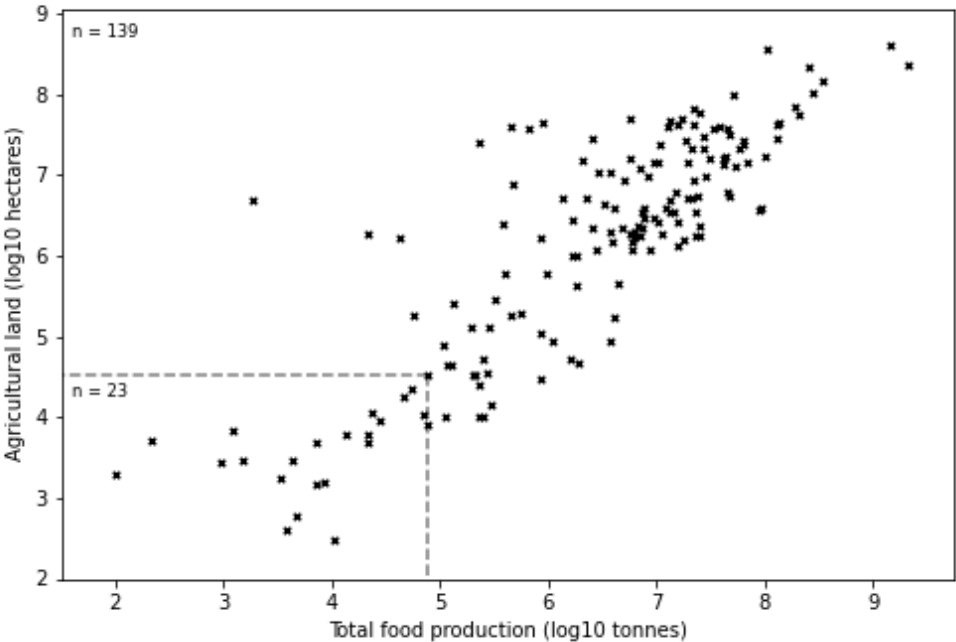

**Figure** A**2. Log of agricultural land area against total food production mass for the year 2017 for all countries in the FAOSTAT database, dissolved states are not included. Countries contained in the small dotted-line box are not included in model processes (n = 23), while the remaining countries are included (n = 139).**


**Appendix B**

**Food commodity groupings**

**Table B1. Vegetal products and grouped vegetal products. Grouped products do not contain any products represented as staple products. The luxury group consists of tea, coffee and cocoa.**

| Individual products | Grouped products |
|---|---|
|  |  |

| | |
|---|---|
| Wheat | Cereals |
| Rice | Fruits |
| Maize | Vegetables |
| Palm Oil | Pulses |
| Rape and mustard seed | Starchy roots |
| Soyabeans | Oil crops |
| Sunflower seed | Spices |
| Potatoes | Sugar crops |
| Cassava | Luxuries |
| Nuts and products | Other vegetal products |


**Table B2. Animal products and groups. In the case of these animal products, the 'individual' animal products represent a small group of products but are dominated by a single product. For ~~example~~example, while *bovine meat* includes derivative products and buffalo, the majority of the bovine meat supply and consumption is formed of cattle meat. There are only two sets of grouped animal products: dairy and 'other meat'. Dairy is a significant contributor to global**
**food supply and demand, but meat products not listed individually do not. Dairy includes milk, butter, ghee and cream. Products such as cheese and yoghurt are also included in the data for milk.**

| Individual products | Grouped products |
|---|---|
| Bovine meat | Other meat |
| Poultry meat | Dairy products |
| Pigmeat | |
| Mutton and goat meat | |
| Eggs | |

**Harvest factors and recovery rates**

**Table B3a. Vegetal product harvest factors – the ratio of the mass of useful product to above ground biomass. Values**
**in this table are adapted from ~~from~~ Krausmann et al., 2008. Where a direct mapping was impossible, the average value of other products was used (for example – vegetables). Fruits are assumed to be permanent crops.**

| | Sub-Saharan Africa | North Africa and | Europe | Central and | East and South- | Oceania | North America | Latin America |
|---|---|---|---|---|---|---|---|---|

| | | West Asia | | Southern Asia | East Asia | | | |
|---|---|---|---|---|---|---|---|---|
| Wheat | 2.3 | 1.5 | 1.3 | 1.7 | 1.5 | 1.2 | 1.2 | 1.5 |
| Maize | 3.5 | 3 | 1.6 | 3.5 | 3 | 1.2 | 1.2 | 3 |
| Rice | 1.5 | 1.2 | 1.2 | 1.5 | 1 | 1.2 | 1.2 | 1.2 |
| Soyabeans | 1.5 | 1.5 | 1.4 | 1.5 | 1.2 | 1.2 | 1.2 | 1.5 |
| Potatoes | 1 | 1 | 1 | 1 | 1 | 1 | 1 | 1 |
| Nuts | 1.5 | 1.5 | 1.2 | 1.5 | 1.2 | 1.2 | 1.2 | 1.5 |
| Cassava | 0.8 | 0.8 | 0.8 | 0.8 | 0.8 | 0.8 | 0.8 | 0.8 |
| Rape and ~~mustardseed~~mustard seed | 2.3 | 2.3 | 1.9 | 2.3 | 2.3 | 1.9 | 1.9 | 2.3 |
| Palm oil | 1.9 | 1.9 | 1.9 | 1.9 | 1.5 | 1.9 | 1.9 | 1.9 |
| Sunflower seed | 2.3 | 2.3 | 1.9 | 2.3 | 2.3 | 1.9 | 1.9 | 2.3 |
| Cereals | 2.3 | 1.5 | 1.25 | 1.7 | 1.5 | 1.2 | 1.2 | 1.5 |
| Oil crops | 2.3 | 2.3 | 1.9 | 2.3 | 2.3 | 1.9 | 1.9 | 2.3 |
| Pulses | 0.4 | 0.4 | 1 | 0.4 | 0.4 | 1 | 1 | 0.4 |
| Starchy roots | 1 | 1 | 1 | 1 | 1 | 1 | 1 | 1 |
| Sugar crops | 0.7 | 0.7 | 0.5 | 0.7 | 0.7 | 0.7 | 0.5 | 0.7 |
| Fruits | 2.5 | 2.5 | 2.5 | 2.5 | 2.5 | 2.5 | 2.5 | 2.5 |
| Vegetables | 1.9 | 1.5 | 1.2 | 1.6 | 1.5 | 1.3 | 1.3 | 1.6 |
| Spices | 1.1 | 1.1 | 1.1 | 1.1 | 1.1 | 1.1 | 1.1 | 1.1 |
| Luxuries | 0.5 | 0.5 | 0.5 | 0.5 | 0.5 | 0.5 | 0.5 | 0.5 |
| Other vegetal products | 1.9 | 1.5 | 1.3 | 1.6 | 1.5 | 1.3 | 1.3 | 1.5 |

**Table B3b. Vegetable product residue recovery factors – the recovered proportion of potential harvest residues. As with table B3a, this table is also adapted from Krausmann et al., 2008.**

| | Sub-Saharan Africa | North Africa and | East Europe | West Europe | Central and South- | East Asia | North America | Latin America |
|---|---|---|---|---|---|---|---|---|

|  |  | West Asia |  |  | East Asia |  | and Oceania |  |
|---|---|---|---|---|---|---|---|---|
| Cassava and products | 0.8 | 0.8 | 0.3 | 0.0 | 0.8 | 0.8 | 0.0 | 0.8 |
| Cereals - Excluding Beer | 0.9 | 0.8 | 0.8 | 0.7 | 0.9 | 0.8 | 0.7 | 0.8 |
| Fruits - Excluding Wine | 0.8 | 0.7 | 0.6 | 0.4 | 0.8 | 0.7 | 0.4 | 0.7 |
| Luxuries (excluding Alcohol) | 0.8 | 0.7 | 0.6 | 0.4 | 0.8 | 0.7 | 0.4 | 0.7 |
| Maize and products | 0.9 | 0.8 | 0.8 | 0.7 | 0.9 | 0.8 | 0.7 | 0.8 |
| Oilcrops | 0.9 | 0.8 | 0.8 | 0.7 | 0.9 | 0.8 | 0.7 | 0.8 |
| Other | 0.8 | 0.7 | 0.6 | 0.4 | 0.8 | 0.7 | 0.4 | 0.7 |
| Palm Oil | 0.9 | 0.9 | 0.9 | 0.9 | 0.9 | 0.9 | 0.9 | 0.9 |
| Potatoes and products | 0.8 | 0.8 | 0.3 | 0.0 | 0.8 | 0.8 | 0.0 | 0.8 |
| Nuts and products | 0.9 | 0.8 | 0.8 | 0.7 | 0.9 | 0.8 | 0.7 | 0.8 |
| Pulses | 0.5 | 0.5 | 0.5 | 0.0 | 0.5 | 0.5 | 0.0 | 0.5 |
| Rape and ~~Mustardseed~~Mustard seed | 0.7 | 0.7 | 0.7 | 0.7 | 0.7 | 0.7 | 0.7 | 0.7 |
| Rice (Milled Equivalent) | 0.9 | 0.8 | 0.8 | 0.7 | 0.9 | 0.8 | 0.7 | 0.8 |
| Soyabeans | 0.5 | 0.5 | 0.5 | 0.0 | 0.5 | 0.5 | 0.0 | 0.5 |
| Spices | 0.8 | 0.7 | 0.6 | 0.4 | 0.8 | 0.7 | 0.4 | 0.7 |
| Starchy Roots | 0.8 | 0.8 | 0.3 | 0.0 | 0.8 | 0.8 | 0.0 | 0.8 |
| Sugar Crops | 0.8 | 0.8 | 0.3 | 0.0 | 0.8 | 0.8 | 0.0 | 0.8 |
| Sugar cane | 0.9 | 0.9 | 0.9 | 0.9 | 0.9 | 0.9 | 0.9 | 0.9 |
| Sunflower seed | 0.5 | 0.5 | 0.5 | 0.5 | 0.5 | 0.5 | 0.5 | 0.5 |
| Vegetable Oils | 0.8 | 0.7 | 0.6 | 0.4 | 0.8 | 0.7 | 0.4 | 0.7 |
| Vegetables | 0.8 | 0.7 | 0.6 | 0.4 | 0.8 | 0.7 | 0.4 | 0.7 |
| Wheat and products | 0.9 | 0.8 | 0.8 | 0.7 | 0.9 | 0.8 | 0.7 | 0.8 |


**Appendix C**

**Table C1. Maximum portion ($z$) of livestock feed that can be derived from each residue source.** These values are taken from FALAFEL (Powell, 2015).

| Livestock Product | Harvest residues | Processing waste | Post-production waste |
|---|---|---|---|

| | | | |
|---|---|---|---|
| Dairy | 25% | 5% | 0% |
| Bovine Meat | 25% | 5% | 0% |
| Eggs | 0% | 11% | 0% |
| Poultry Meat | 0% | 11% | 0% |
| Pigmeat | 5% | 15% | 45% |
| Mutton & Goat Meat | 20% | 11% | 0% |
| Other Meat | 20% | 5% | 0% |

**Appendix D**

**Table D1. Inputs, values and data used to produce the anchor scenario in C-LLAMA.**

| Input | | Value or data | | Source |
|---|---|---|---|---|
| Population | | SSP2 population trajectory | | Fricko et al., 2017. |
| Idealised food supply target calories | | 3200 (kcal/capita/day) | | Kearney, 2010; Alexander et al., 2017; Parfitt et al., 2010 |
| Idealised food supply target year | | 2100 | | Aligns with Paris agreement temperature goals. |
| Overall efficiency improvement | | 0.0 | | There is no enforced change to overall agricultural efficiency in the anchor scenario. |
| Change to vegetal diet | | 0.0 | | No enforced change to portion of food energy from vegetal products in the anchor scenario. |
| Change to dairy diet | | 0.0 | | No enforced change to portion of food energy from dairy products in the anchor scenario. |
| Waste factor limits | | *Subsistence* | *Industrial* | Refer to section 4.2 and Table 2. |
| | *Post production* | 0.07 | 0.3 | |

| | | | |
|---|---|---|---|
| *Processing* | 0.10 | 0.06 | |
| *Distribution* | 0.5 | 0.05 | |
| Post-production waste to feed | 0.40 | 0.05 | (Powell, 2015) |
| Other waste to feed | 0.15 | 0.40 | |

## Appendix E

**Table E1. Table of aggregated land-use areas at a regional level in the C-LLAMA anchor scenario. Values are in hectares.**

| Region | Item | 2020 | 2030 | 2040 | 2050 |
|---|---|---|---|---|---|
| NORTHERNAMERICA | Food Crops | 1.44E+08 | 1.48E+08 | 1.52E+08 | 1.55E+08 |
| | Pasture | 2.78E+08 | 2.83E+08 | 2.88E+08 | 2.9E+08 |
| | Fodder Crops | 59102644 | 67120616 | 74375777 | 80618936 |
| SOUTHAMERICA | Food Crops | 92609372 | 93982867 | 95152719 | 96217646 |
| | Pasture | 4.53E+08 | 4.77E+08 | 4.98E+08 | 5.19E+08 |
| | Fodder Crops | 41950885 | 47980106 | 53268586 | 57606791 |
| CENTRALAMERICA | Food Crops | 24486324 | 24883564 | 25113116 | 25144180 |
| | Pasture | 90549873 | 89083781 | 87279927 | 84992096 |
| | Fodder Crops | 10298992 | 10854346 | 11290526 | 11608143 |
| CARIBBEAN | Food Crops | 5701552 | 5814101 | 5898041 | 5947747 |
| | Pasture | 4735629 | 4983965 | 5261775 | 5484907 |
| | Fodder Crops | 768887.3 | 867706.7 | 962794.9 | 1046271 |
| EASTERNAFRICA | Food Crops | 44627557 | 44787338 | 44348960 | 43227455 |
| | Pasture | 1.96E+08 | 2.12E+08 | 2.26E+08 | 2.37E+08 |
| | Fodder Crops | 27711651 | 30686740 | 33367389 | 35708687 |
| WESTERNAFRICA | Food Crops | 70371563 | 70589754 | 69519405 | 66931695 |
| | Pasture | 1.87E+08 | 1.95E+08 | 2E+08 | 2.01E+08 |
| | Fodder Crops | 33461679 | 34978667 | 36429016 | 37396721 |
| NORTHERNAFRICA | Food Crops | 33026116 | 33370613 | 33414347 | 33119937 |
| | Pasture | 60159833 | 64754843 | 68517651 | 72314640 |
| | Fodder Crops | 13791410 | 14023182 | 14217248 | 14445584 |
| SOUTHERNAFRICA | Food Crops | 9524790 | 9679176 | 9798009 | 9876671 |
| | Pasture | 1.52E+08 | 1.6E+08 | 1.66E+08 | 1.7E+08 |
| | Fodder Crops | 4506502 | 4753946 | 4954572 | 5111965 |
| MIDDLEAFRICA | Food Crops | 13486305 | 13360842 | 12961523 | 12276739 |
| | Pasture | 1.22E+08 | 1.29E+08 | 1.34E+08 | 1.37E+08 |

| | | | | | |
|---|---|---|---|---|---|
| | Fodder Crops | 7890880 | 8567387 | 9229300 | 9799051 |
| CENTRALASIA | Food Crops | 20611646 | 20265557 | 19786873 | 19078910 |
| | Pasture | 2.62E+08 | 2.91E+08 | 3.16E+08 | 3.37E+08 |
| | Fodder Crops | 18174812 | 20534064 | 22860266 | 25172668 |
| EASTERNASIA | Food Crops | 1.09E+08 | 1.1E+08 | 1.1E+08 | 1.09E+08 |
| | Pasture | 5.15E+08 | 5.44E+08 | 5.64E+08 | 5.98E+08 |
| | Fodder Crops | 37005906 | 39669148 | 41686753 | 43263691 |
| SOUTHEASTERNASIA | Food Crops | 1.09E+08 | 1.11E+08 | 1.12E+08 | 1.12E+08 |
| | Pasture | 17297852 | 18576964 | 19557564 | 20315505 |
| | Fodder Crops | 14473651 | 15505407 | 16259165 | 16769792 |
| SOUTHERNASIA | Food Crops | 1.97E+08 | 1.96E+08 | 1.93E+08 | 1.87E+08 |
| | Pasture | 79168805 | 84059651 | 88382231 | 91860597 |
| | Fodder Crops | 43910757 | 47504797 | 50725443 | 53662898 |
| WESTERNASIA | Food Crops | 27309881 | 27202184 | 26859498 | 26261145 |
| | Pasture | 2.2E+08 | 2.21E+08 | 2.21E+08 | 2.18E+08 |
| | Fodder Crops | 11275983 | 11807883 | 12254763 | 12668759 |
| EASTERNEUROPE | Food Crops | 1.47E+08 | 1.45E+08 | 1.43E+08 | 1.39E+08 |
| | Pasture | 1.22E+08 | 1.35E+08 | 1.48E+08 | 1.59E+08 |
| | Fodder Crops | 51319529 | 56769933 | 62024203 | 67294659 |
| WESTERNEUROPE | Food Crops | 26417618 | 26343260 | 26198250 | 25928975 |
| | Pasture | 25054210 | 27539612 | 29787514 | 31851160 |
| | Fodder Crops | 8910851 | 9151293 | 9307426 | 9462245 |
| NORTHERNEUROPE | Food Crops | 7865396 | 7719001 | 7647723 | 7594364 |
| | Pasture | 24317404 | 28344686 | 32765160 | 35600568 |
| | Fodder Crops | 10971461 | 11199462 | 11375065 | 11577335 |
| SOUTHERNEUROPE | Food Crops | 30710974 | 30774176 | 30477722 | 29798789 |
| | Pasture | 25655657 | 28352163 | 30802283 | 32903614 |
| | Fodder Crops | 6694103 | 7164839 | 7548375 | 7914750 |
| AUSTRALIAANDNEWZEALAND | Food Crops | 18777577 | 18029310 | 16918547 | 15466293 |
| | Pasture | 3.5E+08 | 3.44E+08 | 3.37E+08 | 3.27E+08 |
| | Fodder Crops | 13146533 | 14248087 | 15188036 | 16059053 |
| POLYNESIA | Food Crops | 22455.07 | 23645.13 | 24856.15 | 26018.59 |
| | Pasture | 25840.88 | 28426.17 | 30704.33 | 32588.92 |
| | Fodder Crops | 33490.12 | 34066.78 | 34799.68 | 36579.62 |
| MELANESIA | Food Crops | 538313.8 | 549717.5 | 560089.8 | 568835.8 |
| | Pasture | 405376.9 | 420432.4 | 431176.1 | 437022.8 |
| | Fodder Crops | 20676.97 | 21164.86 | 21608.33 | 22026.5 |

 **Appendix F**

Counter-intuitive behaviour arises when setting the proportion of animals fed through fodder and residues (fed without forage - FWF) to extreme values. Decreasing the FWF factor (more animals are fed through pasture) leads to an increase in land-use by 2050. This is expected, as pasture is typically far less land-efficient than housed animals fed through fodder and residues (Pikaar et al., 2018). However, this trend does not continue when the FWF is increased, instead an increased land-use is observed. The behaviour of the FWF prompted further investigation; the factor was scaled by a range of values between 0.5 and 1.5 to observe the behaviour around the default values (a scaling of 1.0), the global agricultural land-use values for which are shown in Fig. C1.

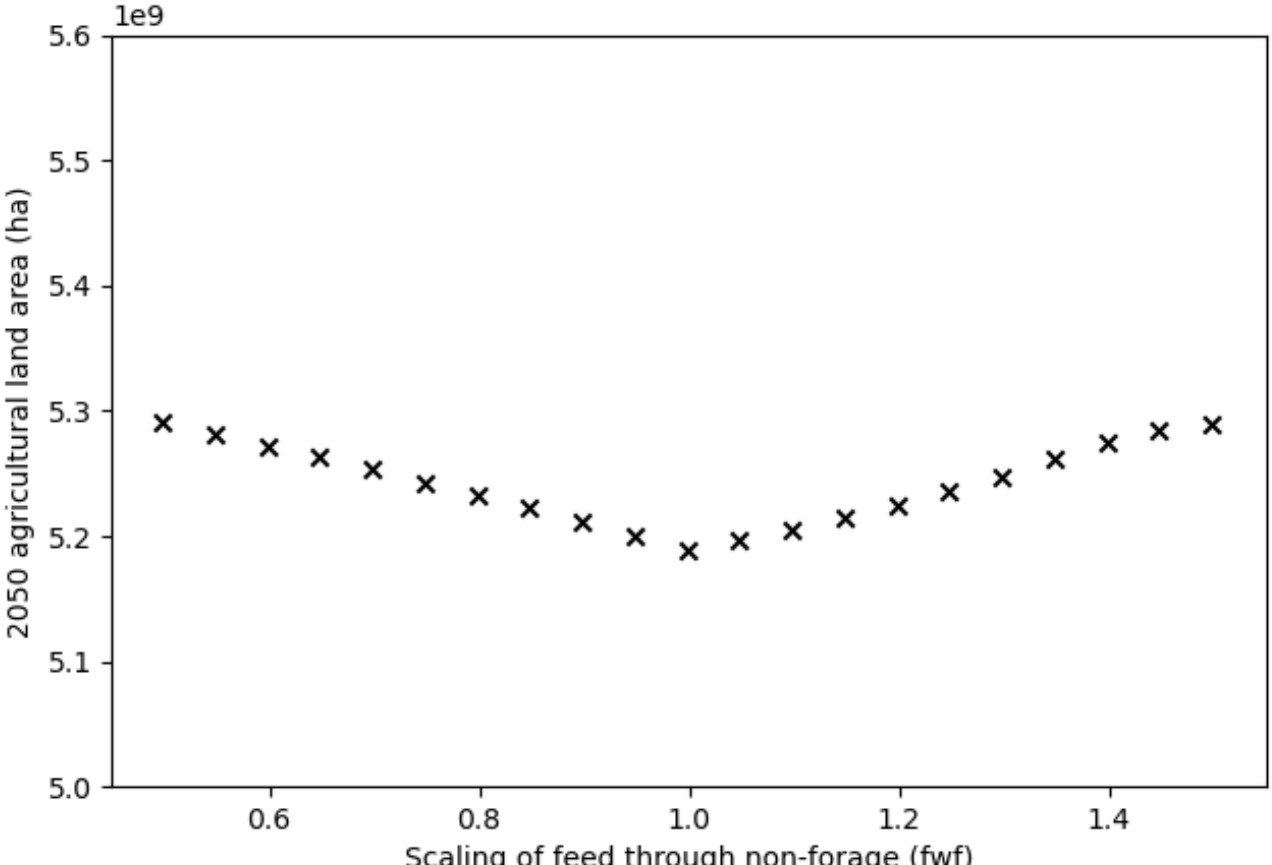

**Figure F1. Change in global agricultural land-use when varying the proportion of livestock feed from non-forage (FWF).**

Inspection of the land-use for pasture, fodder and food crops revealed that food crop land-use was constant as expected since only animal product production methods are being varied. Fodder crop land-use also behaved as expected – increasing with

FWF, as more fodder crops must be produced to meet the feed demand of animals not produced on pasture. However, pasture
did not behave as expected, instead following the same trend as the global land-use, with an increased land-use when varying
the FWF factor in either direction. The cause of this behaviour has been identified as the scaling method applied to pasture
land area. When the scaling is turned off, variations in the FWF factor lead to expected behaviour: global land use decreases
as FWF increases. The effective pasture yield is calculated using the projected 2017 land-use value before any scaling is
applied. When FWF is increased the quantity of animal products produced on pasture decreases, including the 2017 value,
however the historical pasture area remains unchanged. The result is an artificial decrease in effective pasture yield as FWF
increases when the scaling is applied, as shown in Fig. C2.

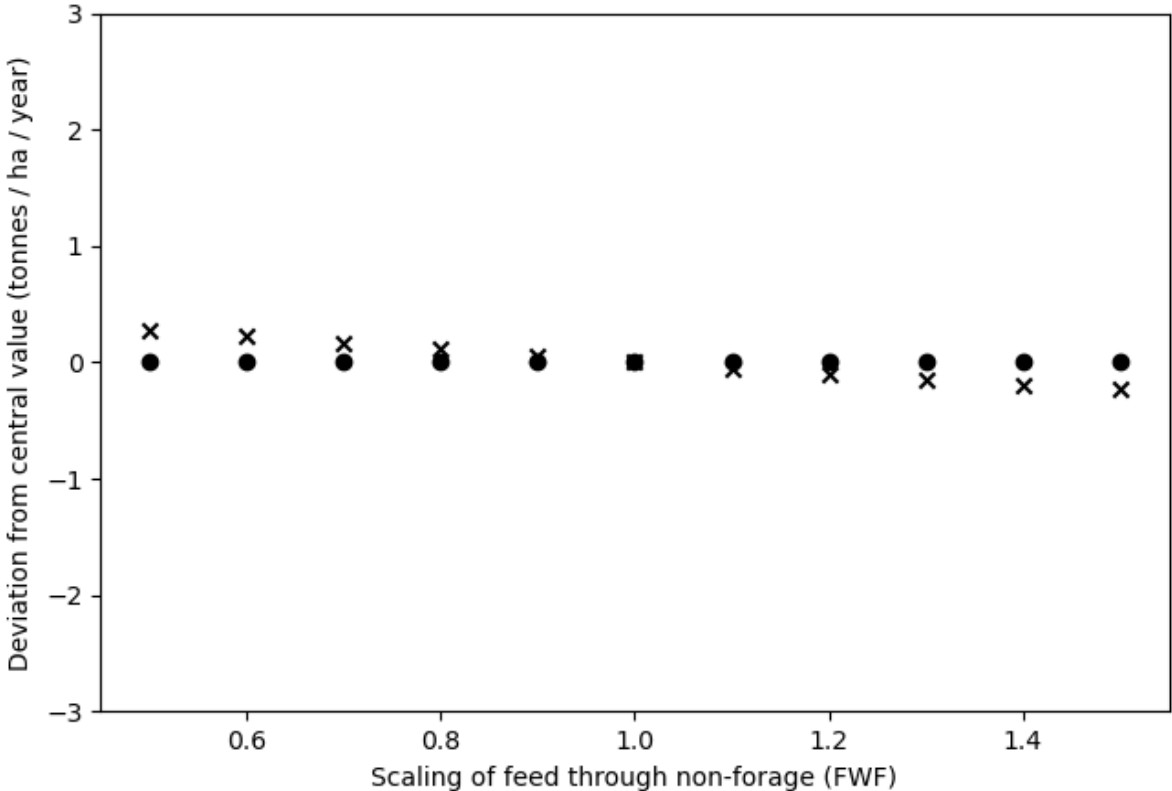

**Figure F2. Magnitude of change in pre and post global mean scaled effective pasture yield for forced scaling of livestock feed through non forage (FWF).**

To resolve this and any similar anomalies arising from scaling methods, the effective pasture yield is now scaled based on the
projected pasture area in the anchor scenario, regardless of the scenario parameters. This can introduce minor discrepancies in
the early years of the projection when setting factors to a fixed value, but this is not the normal mode of operation for the
model. This sensitivity test varied the FWF factor for the entire projection, including the starting values, where in normal
model operation any changes to this factor would be applied as a gradual deviation from the normal value. For example, the
scaling might vary from 1.0 in 2017 to 1.5 in 2050, as opposed to being 1.5 from the start as in this sensitivity analysis.

## Code availability

C-LLAMA model source code can be found at 10.5281/zenodo.5083000.

## Author contributions

TB developed C-LLAMA v1.0 and performed model runs. TB prepared the original manuscript, all co-authors contributed to
manuscript review.

## Competing interests

The authors declare that they have no conflict of interest.

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
