# Peer review of "C-LLAMA 1.0: a traceable model for food, agriculture, and land-use"

_Geoscientific Model Development, 2021_

## Author Comment (AC1)

**Author response to reviewer comments**

The authors thank the reviewers for their comments. We are happy to learn that the reviewers see the value in this work and have evidently taken significant time and effort to assist in the improvement of the manuscript; the level of care taken by all reviewers is evident in the attention to detail of the comments. Several comments noted poor clarification of model processes and details, which have all now been addressed. Your thoughtful suggestions and contributions pertaining to the processes within the model have been carefully considered, with many changes being made to the model as a result, including an updated anchor scenario and a calibration of crop areas. In these instances, all relevant figures and text have been updated.

**Reviewer 1**

This paper presents a relatively simple non-economic model representing the food system and its relation to agricultural production and land use based on empirical data from the FAO. While the ambition is interesting and laudable the manuscript needs major improvement before publication. Also, I have major concerns with several key assumptions made in the modelling.

**General comments**

R1.1 - The introduction is very poor. The specific models that are referred to as 'opaque' are not discussed in any detail. These models are the agro-economic components of the IAMs, specifically GLOBIOM, MAgPIE, GCAM, AIM, IMAGE. While these models might be considered relatively complex, they are described in high detail in numerous publications and model intercomparisons and are based on fairly basic economic principles. Moreover, some of these models by now are open source available (GCAM https://github.com/JGCRI/gcam-core/releases and MAgPIE https://github.com/magpiemodel) or have detailed model descriptions online (IMAGE https://models.pbl.nl/image), and for all models large amount of results are available for public use (https://data.ene.iiasa.ac.at/iamc-1.5c-explorer). The JULES and LPJmL models that are discussed explicitly are a very different type of model that do not consider the food system and are therefore irrelevant for this discussion. A key improvement for this paper needs to be a proper discussion of why the type of model presented her is of relevance and interest, especially in relation to models describing the food system in the literature. Currently the only explanation for the endeavour undertaken here is that these are opaque, which is not sufficient. In addition, an argumentation is needed why a move from a simple global approach to a country-based approach is an improvement as it inevitably increases the complexity of the model. Currently there is only an explanation of the advantage of moving from excel to python which is not so relevant for this manuscript in my view (though definitely an improvement).

Response – Thank you for this comment, we agree that the introduction had room for improvement and have taken steps to address this, several sentences have been added contextualising the usefulness of C-LLAMA in relation to IAMs. [Lines 24 - 34]

R1.2 - Throughout the manuscript, there is extensive reference to the FALAFEL model which is not described itself. Also, in the results section comparisons to results from the FALAFEL model are presented. If a model comparison is made, the model to which it is compared also needs to be presented and explained in detail. I think however that it is preferable and more clear if the paper only discusses the C-LLAMA model and does not lean too much on results from an older model. On the other hand, the paper would benefit from more extensive discussion of the results in light of food system trends in the recent past: do the projected developments of land use, food and feed production and per capita consumption make sense compared to real-world trends of the last twenty to thirty years?

Response – Thank you for your comment, C-LLAMA acts as a successor to FALAFEL; the core model processes are identical, with additional components and complexity required to make the transition to a country-level model. Given that FALAFEL is published work (Powell 2015; Powell & Lenton 2012 10.1039/C2EE21592F), the reason for the comparison was to 'sense check' the results of C-LLAMA and confirm that the model functions as expected.

R1.3 - The agricultural industrialisation parameter is based on food energy consumption per capita. This is surprising as this variable is very indirectly related to agricultural industrialisation. Indeed, highly modernized countries such as Korea and Japan show very different consumption for cultural and physical reasons that are completely unrelated to farming practices. On the other hand, middle income countries such as Morocco, Algeria, Turkey and Romania hive higher consumption than high income countries such as Netherlands and Australia, while the former definitely have less industrialized farming than the latter. There are variables available that are more directly linked to industrialization. I would recommend to look into yield gap data, for example from the

yield gap atlas (https://www.yieldgap.org/, no global coverage unfortunately) or from analyses such as Mueller et al 2012 (https://www-nature-com/articles/nature11420, supplementary data available per country). Also, the industrialisation parameters should definitely be shown in the appendix as it seems to be the dominant parameter as shown by the sensitivity analysis.

Response – We appreciate that the choice to use food-supply for this parameter may be a somewhat counter-intuitive one. Based on comments from both reviewers, we realise that the choice of name for the parameter is quite poor and confusing, and thus have decided to re-name it to 'food-system efficiency parameter'. Throughout the model processes, the metric is never used to make any predictions or assumptions surrounding yield or agricultural practices (these are accounted for in yield and harvest residue data); instead it is used to infer consumer (and commercial) behaviour and food losses due to food system infrastructure. Many different approaches including on-the-ground industrialisation metrics such as machinery, arable land per capita, irrigation and fertiliser usage (data.worldbank.org/indicator), along with other approaches such as GDP (WorldBank) were analysed as potential indicators of food system efficiency. Machinery, arable land, irrigation and fertiliser data were incomplete or outdated for many countries. Rather than reflecting food system efficiency, these data are heavily influenced by dominate crop types and climate variations, drowning out any underlying influence from food system behaviours and infrastructure. Yield gap data was also discounted, given that the parameter isn't used to inform any values surrounding yield or agricultural practices; we feel that the change of parameter name clarifies the reason for this decision. The primary issue with GDP is the very high level of disparity at all GDP levels (for example, Switzerland has 150% the GDP of the UK, yet their food-systems are very similar). The figure below illustrates the fact that cereal yields and GDP both vary dramatically even between countries of similar development. Initially the UNEP food sustainability report (www.unep.org/resources/report/unep-food-waste-index-report-2021) was promising, but the data is fairly incomplete (52 countries total), with several regions having no data points. We concluded that our food-system efficiency metric provides a sound method for the approximation of food-waste habits within a country given the process resolution of this model and in the absence of a coherent dataset of food-waste estimates for all countries. Values estimated on a case-by-case basis would be preferable, however such an approach was outside the scope of this piece of research; such an approach would be a high priority for inclusions in later versions of the model.

[Figure]

R1.4 - The model does not take into account other use of crops. Most notably bioenergy is crucial, with nearly half of all maize production in the USA and nearly half of sugarcane in Brazil used for bioenergy. Also cotton production for clothing is important.

Response – Thank you for your comment, it has highlighted an omission from the manuscript made in error. The two industrial uses of primary crops are taken into account as far as production data goes: a factor is applied to each of these to reduce the production value to a 'food-only' estimate. Regarding non-food crops, we appreciate that their impact on land and resource use is significant, however their addition to this model is outside the scope of this work: the main goal of the model is to explore the impacts of changes within the food system on land-use.

In future iterations of the model the inclusion of non-food crops, including bioenergy feedstock, will be a high priority inclusion. Sentences have been added to describe the reduction factor used to estimate 'food-only' use for maize in the USA and sugarcane in Brazil (as the two dominant cases for industrial use of these crops), along with additional references for the values used. [Lines 224 – 227]

R1.5 - The assumption that all countries converge to 3200 kcal per capita by 2100 is crucial for total production estimates. While such an assumption is reasonable in stylized scenarios assessing the effects of dietary change, I don't believe it makes sense for business-as-usual projections. In the recent past consumption in countries above the 3200 kcal thresholds have shown no substantial decreases in consumption, so this would be a clear break of the historical trend. On the other hand, in countries that for non-economic reasons have relatively low consumption (e.g. earlier discussed Japan and Korea) there is no reason to assume an increase to 3200 kcals.
Response – Thank you for your comment, we completely agree that the convergence of food supply to 3200 kcal/cap/year is not representative of a business-as-usual scenario and an oversight on our part, we have therefore replaced this prescribed trajectory with a linear projection from the historic values food supply, as is done in other aspects of the model. The 'anchor' scenario has been updated: linear projections of historical total food supply for a given country are now made, as opposed to forcing a trajectory toward a global value. All relevant plots have been updated, the text describing the production of the food-supply trajectories has been updated to reflect the changed method. [Lines 104 – 111; Figures 2, 4-6, F1, F2]

R1.6 - In the second sentence of section 3.5.1 it is explained that the land use area results are projected based from the most recent FAOSTAT value. I find this very odd and it is unclear to me how much effect it could have. Are resulting land use areas from the model very different from the FAO reported values in some regions? This could indicate important mismatches in the modelling and I would like to see these data. And why don't you choose to use a calibration factor? In my view this would not violate the purpose of the model to explore sensitivities (a goal which should be mentioned in the introduction if it is indeed a key purpose of the model).
Response – Thank you for your comment, the changes made here as a result have improved the validity of the model projections greatly. In the original version, linear projections made in the model were not calibrated to the historical data, leading to small 'kinks' at the start of many of the projected values compared to the historic. To remove these kinks, the first projected value was aligned with the historic value at the land-use stage.  This was done at the final stage to avoid compounding any errors that might be incurred by doing this several times throughout the model processes. Regarding the calibration factor – this is the method that was used for the pasture effective yield calculations but not for land-use, which was instead done by simply matching the first projected value to the last historic value. A calibration factor is a far more robust method and thus in the resubmitted version the alignment method has been replaced with the calibration method. Sentences have been added to indicate the use of a calibration factor for pasture and crop land-use areas [329].

R1.7 - The results section only presents data on per capita consumption and land use. I would be especially interested in total required production of food and feed crops, fodder, yields and livestock efficiencies. To be able to understand the model dynamics it is quite important to add these. Also all results should be presented at the minimum at the regional level (and if possible even country level in appendix).
Response – Thank you for your comment. Regional land-use data is now presented in appendix E. While we appreciate that production data is important, since the calibration is only carried out at the end of the model processes (at the land-use stage, see R1.6), we decide not to include this in the manuscript. We note however, that yield, production demand/allocation, livestock yields are output as intermediate serialised objects from the model, so can be accessed directly by users. [Line 550]

**Detailed comments**
**Introduction**
R1.8 - Line 16-18: these statements need references. I would maybe refrain from discussing mitigation strategies such as afforestation and BECCS in the introduction. This article is about the food system and resulting land use: the GHG emissions of the agriculture itself and land-use change, as well as other environmental impacts are more relevant in that context.
Response – Thanks for this comment, four references have been added for these statements. In the context of a model description paper introduction, we agree that these mitigation strategies might seem out of context, however we feel that it is important to highlight the topical importance of C-LLAMA and land-use modelling in general in the context of these issues. [Line 19]

R1.9 - Line 25-26: 'United Nations 2015' reference is not available in the reference list, if this is about the Paris agreement I think it should refer to the UNFCCC.

Change – This reference has been replaced with the more appropriate UNFCCC reference.

R1.10 Line 26: I was taught that 'land-use' only needs a dash when used as ad adverb, e.g. in land-use change, but not when it is used as a noun. However, I am not a native speaker so not sure, but please check.

Response – The hyphen has been removed in this instance and care has been taken to use the correct punctuation throughout the remainder of the manuscript.

R1.11 - Line 27: the statement 'somewhat opaque' is very vague and not sufficient as underpinning of the added value of this model (see also general comments).

Response – See response to R1.1.

R1.12 - Line 37-40: I don't know the exact policy of the GMD journal, but in my view the programming language used does not require lengthy discussion in a scientific article. Also it is quite self-evident that python is advantageous over Microsoft excel. The fact that your model is open-source is more relevant and can be mentioned as an argument for the transparency of your approach.

Response – Thanks for this comment, we have removed reference to the advantages of Python over Excel and have now made reference to the open-source code and data. A sentence referring to the advantages of Python has been added along with a sentence referring to the open-source nature of the model and the public availability of all inputs. [Line 43]

***Model overview***

R1.13 - Line 50: a normal ref to the FAOSTAT website is sufficient here.

Change – The reference has been updated to the correct FAOSTAT web page.

R1.14 - Table 1: the names of the program modules is not so relevant for a journal in my view. It is important to provide the model open-source including a documentation of the different modules and how it can be used. You might consider adding it to the appendix, but I don't think it should be part of the main text.

Response – Thank you for this suggestion, the table of program modules has been moved to the appendix which has greatly improved the flow of the manuscript. [Appendix A]

R1.15 - Line 72 and figure 1: it is quite key to know which processes operate at what spatial aggregation. Maybe this can be included in the model structure overview.

Response – Thank you for highlighting the lack of clarity here, two sentences have been added explicitly stating at which level the processes operate, along with the reasoning for this structure [Lines 68 - 72].

R1.16 - Line 73: 'global food production' - I assume you are talking about total agricultural production here, including all crops and animal product consumption. This is definitely not all directly used for food, with crops produced for bioenergy, animal feed, clothing, etc. Please reconsider your terminology.

Response – Thank you for your comment, we realise that the description given here was not explicit enough. Here we specifically refer to food production – non-food crops were removed in the calculations for these plots and values (factors applied to USA and Brazil for non-food use of primary crops maize and sugarcane as described in the response to R1.4). The text 'production mass of crops used for food' has been added to make clear that this does not include non-food uses. [Line 74]

R1.17 - Line 77: 'discounted' - the term discounting in my experience is used in economics to take into account that anything in the future is 'worth less' (simply put), it might be more clear if you use the term 'excluded' here.

Response – The term excluded fits much better in this context, thank you for this suggestion. [Line 77]

R1.18 - Lines 73-95 and figures 2 and 3: a lot of text and figure space is used to explain why a few countries are not considered in the model. In my view this is more appropriate for the appendix and can do with a short description in the main text. The space that is saved on these figures can be used to add additional figures with model results.

Response – We agree that these figures unnecessarily disrupt the flow of the manuscript given the minimal resultant impact on the model processes, so they have been moved to the appendices. [Appendix A]

**Model components**

R1.19 - Line 118: a linear regression is made, but it is unclear of what. What are different variables used? Also it is unclear to me what the effect of this is although it is very important whether people eat relatively more animal or vegetal products. Please present the regressions in the appendix.

Response – Thank you for this comment, several edits have been made to the text to rectify the lack of clarity in this section. All the regressions referred to here are of FAOSTAT data without any modification, so we believe that their inclusion in the appendix is unnecessary. [Lines 104 – 118]

R1.20 - Line 239-241: these yield assumptions could be quite important for the model and are unclear. Does this imply yields in for example Africa can never exceed yields that have already occurred somewhere within the region before, even though most of the continent has relatively high yield gaps? And at what resolution is this analysis done: at the country-level and for each crop? Quite important to add these maximum yields to the appendix.

Response – Thank you for this comment, the changes made as a result have improved the model. We wanted to capture some element of the climatological effects on yield by applying this maxima, however we appreciate that in many regions the maximum achieved yields may fall way below the attainable yield, for this reason we have removed the cap on yield, with a view to addressing this issue using yield-gap data in the future. The cap on yields has been removed in the model code, the anchor scenario has been updated to reflect this and all relevant plots have been updated. [Figures 2, 4-6, F1, F2].

*Model output*

R1.21 - Line 355-356: it is irrelevant to explain that you write out data as a CSV.

Response – This sentence has been removed.

R1.22 - Line 360-361: also the discussion on storage space is irrelevant in my opinion, but maybe GMD has a policy on this that I am not aware of.

Response – The text in this section has been updated, and the reference to storage space removed. [Lines 360, 363]

R1.23 - Figure 4: I am quite surprised about certain trends: the Americas show a strong increase in both crop and pasture land from 2015-2050 while this has not been the case in the last 30 years. The strong increase in pasture in Asia is also odd, as pasture areas have been very stable in the last decades. Can you explain this? If the trends are so different from the recent past, is the model than actually up to the task? Also you might consider excluding the same countries for the historical period as you did in the modelling to make the graphic more clear.

Thank you for your response, this is due to the way trade and land-use work in the model. Regarding trade, as demand for animal products from expanding middle classes in other areas increases, a lot of this demand gets allocated to regions of historical high animal product production (Brazil, China, Argentina, USA etc) in the model. We appreciate that this is not a very good representation of the way trade will work in the real world, but a more complicated system was outside the scope of this work. In future iterations of the model we hope to implement trade matrices or regional trade mechanisms. Regarding land-use, currently there is no accounting for the way that pasture is frequently 'abandoned' in the real world (i.e., rotation or slash/burn practices in many regions), this was initially a goal of the model and is a high priority for future versions of the model. The figure below is included for the reviewer, note the discontinuity in Africa comes from the previously discussed countries not included in the model. The data that underlies this figure is now included as a Table in the appendices. [Table E1]

[Figure]

R1.24 - Figure 5: it would be great to combine this graph with historical data similar to the land use, to see how the trends compare to the recent past.
Response – This figure has been updated to include the historical data. [Figure 3]

R1.25 - Figure 6: the difference in land use expansion between C-LLAMA and FALAFAL is very large, most notably for pasture. Please discuss why this is the case. Also it is surprising that the data are so different in 2015. Why is this?
Response - Thank you for your comment. The reason for these differences are the methods used to estimate pasture in FALAFEL. C-LLAMA derives an empirical 'yield' for pasture, whereas FALAFEL uses estimates of land-productivity and livestock energy uptake. Sentences have been added to discuss this. [Lines 405 - 414]

R1.26 - Figure 7: this figure is very difficult to interpret as the changes are so small. You could consider showing changes over the 2015-2050 period instead.
Response – Thank you for your comment, we elected to remove this figure in light of comments regarding the extensive comparison to FALAFEL (R1.2).

**Discussion**
R1.27 - The discussion requires an additional section that compares the results from C-LLAMA to the wider literature: a lot of models are available that project land use. Please discuss both the type of model and the results in this context.
Response – Thank you for your comment, we agree that the discussion was lacking in context and content and have added paragraphs and a table contextualising the results of C-LLAMA. [Lines 470 – 482, Table 2]

**Reviewer 2**
The authors present an empirical system for extrapolating global historical land use data into the future under user-defined scenario conditions, at the national level. The model appears to mostly capture historical trends and behaves as expected to variations in yield, industrialization, diet, and efficiency. The results are comparable to a global model with a similar structure.
I like the potential application of this model both for data analysis and scenario exploration, but this paper is incomplete in multiple respects. Considerable clarification is needed, including with respect to the knowledge gap that this system aims to fill. My main concerns are summarized here, with specifics following.

*General comments*
R2.1 - Framing and introduction need clarification of why this model is important: what knowledge gap does this fill? One useful aspect is how it summarizes current data and trends. The main aim appears to be able to explore different food demand and supply scenarios and how they may affect land use, in a straightforward manner. This is of importance to the food systems field, especially once the model includes bioenergy crops and afforestation. As it is, it still has value in its simplicity, but I suggest reading and incorporating more food systems literature in order to clarify this model's contribution. The juxtaposition with cmip-style models does not show how this model is a benefit (e.g., why are iams not "traceable?" they have hundreds if not thousands of outputs. how does this

model make it easier to understand land change?). It is also unclear what the point of the comparison with Falafel is.
Response – Thank you for your comment, please see response to R1.1 and R1.2.

R2.2 - Significant clarification of the data and model processes is needed. In many cases I think I understand what you mean because the model needs to work a certain way to achieve your goals, but the description is not sufficient.
Response – Thank you for your comment, it made clear the fact that several sections in the model description were not sufficient. Changes have been made throughout to improve the clarity of description of the data and model processes. These are detailed below in relation to specific comments [e.g. R2.3, R2.8 – R2.32].

R2.3 - There appears to be an inconsistency between food "supply" and calculation of food energy requirement, with respect to whether losses are included and subtracted properly. This may simply be a clarity issue, but it is important to make sure that the operations are consistent with the data represent.
Response – Thank you for pointing out this clarity issue, this has been rectified as addressed in specific comments. Food supply is the food 'reaching' the consumer/commerce, hence it does include post-production waste but not any other kind of waste (processing, distribution etc.). [R2.10, R2.11]

R2.4 - The anchor or baseline scenario should reflect historical trends or conditions so that it can be used for model evaluation, with appropriate sensitivity analyses. It appears that it represents recent historical trends, except in its target calorie requirement, and then some constants are applied or some industrial limits changed for sensitivity. The increase in calories should not be in the anchor and should be a diet change scenario. Further sensitivity analyses are warranted, particularly for the parameters in table B1, and could also be done for the values in tables A3a and A3b.
Response – The authors thank you for your comment, please see response to comment R1.5.

R2.5 - Demonstrating the model's usefulness is important, in line with the reframing indicated above. The increase in target calorie scenario is one example, and I suggest you also add a couple of extreme examples such as full change to vegetal diet and full change to dairy diet. There are some examples in the literature that this model can be compared to.
Response – Thank you for your comment, as above, the reframing has been implemented. A fully vegetarian diet scenario has been added to the efficiency plot. [Figure 6].

*Detailed comments*
*Introduction*
R2.6/2.7 - line 28: Do you mean that the difficulty of interpretation makes it is unclear how different drivers affect land? lines 34-47: How is this related to IAMs and DGVMs? What is the knowledge gap that needs to be filled? Are there advantages of cllama/falafel type models over others? How do cllama and falafel help achieve understanding of land projection and climate targets in addition to other models? Is cllama an extension of falafel? What does the comparison of cllama and falafel tell us?
Response – Thank you for these comments, we realise that DGVMs are not particularly relevant in the context of these simple modelling approaches so this section has been removed. FALAFEL is the progenitor of C-LLAMA and hence the output of C-LLAMA has been heavily compared to that of FALAFEL to 'sense check' the model: the results are very similar which speaks to the validity of the method when moved to a country-level model. Several sentences have been added and removed to better explain the drawbacks with IAMs and contextualise the usefulness of C-LLAMA (and its relevance to FALAFEL). [Lines 24 - 42]

*Model overview*
R2.8 - Need a data section and probably a table that describes all of the data used, including the years. Later you refer to 2017 data but here you state the data are from 1961-2013. Be clear that the model starts in 2018, assuming that the last year of data used is 2013
Response – Thank you for your comment, we appreciate that some of the dates named in the description of various data is confusing. The reason for the mismatch of start and end years for various bits of data is that the 'old' food balance sheets upon which much of the model is based end in 2013 but the rest of the FAO database contains data to 2017. We have updated the text here to more explicitly describe the different datasets used, which we hope precludes the need for a table of used data. [Lines 53 – 56]

R2.9 - Lines 74-78: Here you state that there are 162 countries in the database, but figure 3 shows that 174 countries are included and 21 are not included, which sums to 195 countries. These numbers are not consistent.
Response – Thank you for your comment, this has highlighted an oversight in the figures. To reach the 162 number and produce Figure 2 all countries with zero total food production in 2017 had been dropped, which had not been done in the plotting of Figure 3, hence dissolved states and countries with zero food production mass in 2017 are included in this plot when they should not be. These figures and text have been updated to be consistent, and have also been moved to the appendix following comment from another reviewer. Figure 3 has been updated to be consistent with Figure 2 and the 162 countries mentioned. Figures 2 and 3 have been moved to the appendix. [Figure A1 and Figure A2].

*Model components*
R2.10 - Line 107: More detail is needed for the input data. e.g., this food supply does not include the post-production waste? then is it really food supply or food consumed? It seems more like food demand, either as consumed or consumed + post-production loss.
Response – Thank you for your comment, we appreciate that food-demand is a more accurate description of the quantity being described, however we use the term food supply since it is used in the FOASTAT data and hence has been carried forward into the model itself. 'used' has been replaced with available to improve clarity, text stating the inclusion of post-production waste in this quantity remains. [e.g. Line 98]

R2.11 - Later you base food energy requirement on the "food supply", which seems to have post-production loss included, then subtract three losses: processing, transport, and post-production. Make sure that your F and E and r their FAO source are consistent with each other. For example, FAO data may implicitly exclude all of these losses, which would mean that you are underestimating E and subsequently r because you are adding only post-production loss and then subtracting all three losses. Furthermore, your additional 28% for F appears to be valid only for the developed countries and is too high for subsistence countries (see table 2). Check with section 3.3.1.
Response – Thank you for your comment, this has highlighted a significant clarity issue. Food supply is defined by FAOSTAT as 'food available for human consumption', and so does include post-production waste. Post-production waste is not removed at any point, only used to estimate the waste produced at this stage (that might be used for livestock feed etc). Food supply is food 'reaching' consumers, so in order to reach a 'food produced' quantity (r), the estimated food lost due to processing and distribution are 'added' back on. This also brought attention to a mistake in equation 7 (but not the model code), E should be divided rather than multiplied by the loss factors. The 28% was used only in the context of the previous 'anchor' scenario (see response to R1.5) and is no longer used. Changes have been made to text to improve clarity in food supply (3.1) and food production (3.3) sections. Added a line explaining the use of the post-production loss stream in section 3.2 Corrected multiplication in equation 7. [Lines 98 – 127, 203 – 205, 209 – 214, Equation 7, Table 1]

R2.12 - Lines 109-117: Why is this the default? It seems that the model should use the baseline food "supply" and that a scenario would be to increase this as described.
Response – Thank you for your comment, based on this comment and that of another reviewer, the anchor scenario has been updated and the prescription of a calorie trajectory toward the 2500 + post-production waste has been removed, instead using a linear regression to project total food supply forward for each country (see response to R1.5).

R2.13 - Line 125: What same process? The relative proportion of each commodity within each group?
Response – Yes, this was the intended explanation, but we accept that this was not clear. We have removed 'the same processes' and added a sentence stating that a linear regression is used to project the relative proportions of each commodity within a group. []

R2.14 - Lines 155-156: How does GDP reflect income equality and a high efficiency parameter? How would high GDP represent a majority subsistence agriculture country?
Response – Please see response to R1.3.

R2.15 - Lines 162-166: If high industrialized countries should have a value >1, why is Xa scaled to 0-1? For what year are the results 0.5-1.2? Why is Ftarget multiplied by 0.7 in equation 5?

Response – Thank you for your comment, the oversights made here have been corrected as a result. In the calculation of the parameter, countries that are highly industrialised should come out as >1 as they are 'fully industrialised' and assume the industrialised values for food-waste streams in the model. The 0.5 to 1.2 values are for the most recent historical value (2013 in the 'old methodology' food balance sheets). F_target is multiplied by 0.7 as part of the adjustment process; to normalise values to 0 – 1:

$$z = \frac{x - \min(x)}{\max(x) - \min(x)},$$

equation 5 is an adapted form of this equation (min(x) and max(x) are simply 0.5 and 1.2 respectively). A sentence has been added indicating the years for which the values described are produced. [Lines 152 – 162]

R2.16 - Line 167: If equation 5 is a function of year, how is this projected forward linearly?
Response – Thank you for your comment, as you have noticed, equation 5 is incorrect. F_a should not be a function of year since the parameter is calculated from historical values, this has been addressed and eq 5 is now correct. Equation 5 has been updated; n is now subscripted to reflect the fact that X is not a function of year. [Equation 5]

R2.17 - Lines 178-181: Are there three version of equation 6, for processing, distribution, and post-production, respectively? Based on later descriptions, this factor does not seem to apply to the harvest loss, as this is implicitly captured in the historical trend analysis for crop yield. Later it sounds like this equation is also used for forage and non-forage, which should also be explained here. All factors using this equation should be stated here. waste feed factors also.
Response – Thank you for this comment, the inclusion of 'harvest' in this section was a mistake, this has been updated to correctly read 'processing'. The equation is also used for forage, non-forage and waste-to-feed ratios, text has been added to convey this. [Lines 162 - 169]

R2.18 - Lines 215 and 220: section 3.1?
Change – Now correctly reads section 3.1.

R2.19 - Line 236: What years of data are used?
Change – Added parentheses (2013 to 2017) to indicate the years used. [Line 278]

R2.20 - Line 249: Possibly agricultural greenhouse gas emissions, not global greenhouse gas emissions.
Response – Thank you for your comment, 'the highest contributor' has been replaced with 'one of the highest contributors' and a value range from the referenced work and an additional reference have been added. [Lines 246 - 248]

R2.21 - Lines 265-266: This is confusing because mu is already used, but not in this context. And there is no z in this context, but it is used differently later.
Response – Thank you for this comment, this convention was not clear, hopefully the changes made have improved the clarity sufficiently. Changes made to section 3.2 detailing all uses of the food system efficiency metric, text has been added to section 3.4 stating that the values are informed by the parameter. All instances of mu are for values directly informed by the parameter. [Lines 281 - 285, Table 1]

R2.22 - Lines 270 and 271: Capital Q and year n
Change – Added 'year n' and replaced the lowercase q with a capitalised Q.

R2.23 - Line 287: Does other waste include processing and distribution? be clear here, as above distribution waste is not included in the list. and does this include harvest residue also since it is included as one G in eq 11?
Response – Thank you for your comment, other waste represents processing waste and harvest residues, but not distribution waste since this waste stream is comprised of spoiled and 'lost' food, unable to be retrieved for use as livestock feed. A sentence has been added to clarify the meaning of other waste. [Line 286]

R2.24 - Lines 290-291: What about forage demand? you mention it earlier, but here livestock feed comes only from waste and fodder. What are z? feed source ratios or waste stream proportions? this isn't clear.
Response – Thank you for your comment. The word 'feed' is used here to infer non-foraged food streams for animals. D is the quantity of animal food to come from non-forage (fodder + waste). D contributes to a portion

(mu) of 'Q * FCR', the remainder of which (1 – mu) is to come from forage. The table of potential feed from waste ratios (values of z) referred to was inadvertently not included in the appendices, which has now been addressed. A sentence has been added describing the energy demand for forage-fed livestock product (Q – D). Added a table of z values in the Appendix. [Line 273, Appendix C]

R2.25 - Line 322: Section 3.3
Change – Replaced 4.3 with 3.3.

R2.26 - Line 337: Section 3.4; and some of the livestock land requirement is based on fodder crop area, so this statement is not true.
Response – Thank you for highlighting this clarity issue, the land requirements referred to here are the physical spaces occupied by livestock. The projected areas of pasture and fodder crops are kept separate in the model output. Added parentheses 'in addition to fodder crop production' to reflect this. [Line 336]

R2.27 - Lines 339-340: Shouldn't this already be taken care of by mu-forage and mu-nonforage for pigs and chickens?
Response – Thank you for your comment. mu-forage/non-forage are in relation to the food energy that the animals gain from those two sources, the pasture factor of 0.1 is to reduce their land footprint, as in many regions these animals will forage in and around farm buildings (as opposed to requiring a productive pasture).

R2.28 - Line 344: What pasture land area data are used? What years of data are used overall in this calculation?
Response – Added reference to FAOSTAT land-use data and statement of date-ranges for the data used. [Lines 343 - 345]

R2.29 - Line 347: What is z? z seems to be used for a lot of different things but is not explained or defined clearly in each case. Maybe different variables are needed instead of all of them being z.
Response – Thank you for spotting this mistake, z has been changed to rho in this instance (and the other use of z is more clearly explained). [R2.24, Line 338]

R2.30 - Lines 350 to 352: Combine to calculate the pasture area? The trajectory of what? the historical pasture yield Y or the pasture area? If the yield is scaled, then does pasture area remain constant? Or is the scaling done just at the initial date, and this scaling value stays constant while area changes with changing demand?
Response – Thank you for your comment, this sentence was not clear. 'Combining' has been replaced with 'A simple division' (an equation is not described here as it is a simple division of one quantity by another and I wanted to avoid adding unnecessary complexity). Sentence regarding calibration has been added to improve clarity – the method you describe is the one that was used, the scaling factor was only calculated for 2017 and then used throughout the projection. [Line 352].

R2.31 - Lines 353-353: See comments for appendix c
Response – Please see response to R2.41.

R2.32 - Line 353: Section 4
Change – Corrected section reference.

*Model output*
R2.33 - Be clear that the model starts in 2014, assuming the last year of data used is 2013.
Change – Replaced 'to 2050' with 'from 2017 to 2050'. [Line 356]

R2.34 - I suggest that the anchor scenario be defined to best represent the historical trends in the data, including population and food supply. Then analyses can start from there. SSP2 population may or may not be sufficient for this, and the food supply shouldn't be based on the recent data and not an increase to an idealized amount. Presumably the other middle of the road parameters are based on recent trends. This would present a good empirical analysis of the current state/trajectory based on data, which can then be a basis for other scenarios.
Response – Thank you for your comment, the anchor scenario has been updated as you suggest, please see response to R1.5.

R2.35 - Lines 371-372: Why? if these are extrapolations of historical trends, which they mostly are, then land use trends should not reverse. even the ssp 2 population projections do not diverge that much from recent history over this period. Does this have to do with increasing calories to meet idealized food supply? but this wouldn't apply to Europe.

Response – Thank you for your comment, please see response to R1.23.

R2.36 - Line 385: This is inconsistent with the pasture area increase

Response – The anchor scenario and calibration have been updated, please see R1.5

R2.37 - Section 4.1.1

Change – Corrected section reference.

R2.38 - What is the point of this comparison? Why are the initial areas different?

Response – Thank you for this comment, FALAFEL doesn't make any calibration for initial land-use, instead using historical yield and biomass data to infer land-use. The comparison is made because FALAFEL is the progenitor of C-LLAMA, it is essentially a global version of C-LLAMA in many regards.

*Discussion*

R2.39 - Line 459: Or simple linear extrapolation of historical trends of allocation

Response – Thank you for your comment, this is a great suggestion and we have added a sentence to suggest this. [Line 488]

R2.40 - Lines 474-475: include some example scenarios in this study

Response – Thank you for your comment, a scenario tending to 100% vegetarianism has been added as an example. Two sentences have also been added discussing example of other scenarios that the model could be used to explore. [Figure 6, Line 505]

*Appendix C*

R2.41 - Did you try calculating a scaling factor only once at the initial year, then using this same value throughout? This effectively calibrates the pasture yield to be consistent with history, while allowing pasture area to change independently. scaling throughout conflates pasture yield change and area change. it may make more sense to allow for a separate pasture yield factor that can be set as desired, even by using a trend of historical pasture yield.

Response – Thank you for your comment, this method was explored, but we realised that it did not allow for the effective 'intensification' of pasture that is occurring in many regions globally.

---

## Author Response (AR2)

**C-LLAMA 1.0: a traceable model for food, agriculture, and land-use**

Author comment: We thank the reviewer for their comments on the revised manuscript. As with the previous iterations of feedback it is abundantly clear that you have taken significant time and effort to assist in the improvement of this manuscript and for that we are very grateful. We have attempted to address all the points raised although given the relatively short timeframe in which to make these changes their extent is less than the previous revisions.

**Overall response**

This is a revision of a previously reviewed manuscript. The authors have made extensive revisions in response to reviews and have improved the paper. However, there are still some issues that need to be addressed prior to publication.

1) The framing is still incomplete. What gap does this model fill? The IAM discussion is not very relevant as the goal of this model is stated as being different than the goals of IAMs. I suggest looking at the model goal and the potential model uses in the discussion and then developing an introduction that shows how this model is needed to address such questions (which may or may not involve pointing out how IAMs are inadequate for the task).

A thorough food system science literature review should help with this framing by providing the background and impetus for this type of model. Furthermore, there are other simplified models out there to discuss and compare to. For example, check out the SIMPLE model:

https://www.gtap.agecon.purdue.edu/resources/res\_display.asp?RecordID=4021

https://www.sciencedirect.com/science/article/abs/pii/S1364815220304205

2) Further clarification is needed, and some equations need to be checked, particularly for livestock. See details below.

3) I also suggest showing additional results of the examples and discussing the implications. This helps demonstrate the usefulness of the model in the context of the gap it is aiming to fill.

**Specific comments and suggestions**

**Abstract**

What are some key findings of the evaluation and sensitivity analyses?

e.g., the model behaves as expected under historical extrapolation, under the sensitivity analysis, and under a vegetarian scenario that reduces land use area

AR: Thank you for your comment, we have added to the abstract to convey these points.

**Introduction**

line 17: "afforestation"

AR: This typo has been corrected.

lines 32-34: You have not yet introduced c-llama. I suggest making a more general statement regarding the value of simpler, more confined models (preferably with a reference). There is a model called SIMPLE originally developed by Uris Baldos and Thomas Hertel that may be relevant here.

*AR:* Thank you for your comment, the premature reference to C-LLAMA has been removed and replaced with text regarding the general applicability of simple models.

line 38: spiritual successor? do you mean spatial successor?

*AR:* Thank you for your comment, this turn of phrase is perhaps not applicable here and has been removed.

**Food system efficiency**

line 156: can you give a qualitative definition of Ftarget here? otherwise it is difficult to fully understand the parameter and also the following explanation for Japan and Korea having low values.

Actually, I didn't find the definition in section 4.1

AR: Thank you for your comment, this was an oversight when changing the anchor scenario to project food supply rather than prescribe it. A brief description of F\_target has been added with two references.

line 160: "efficiency" in place of "industrialization"?

AR: Industrialisation has been replaced with efficiency.

**Food production**

line 212, eq 7: I think the denominator should be the product of 1-mu

AR: this is correct, thank you for pointing this out. (This is correct in the code!).

**Livestock**

lines 257-258: these numbers appear to be feed efficiency, which is the inverse of FCR

lines 265-268 (eq 10): the feed quantity is higher than the output production quantity. Make sure you are actually using FCR and not feed efficiency.

lines 268-271: clarification needed: I assume you are talking about mu-non-forage here.

*AR:* Thank you for these comments. I was incorrectly describing feed efficiency as FCR as you point out, which has now been corrected. Eq 10 has also been updated to reflect this.

lines 273-275: due to the FCR, this is not a valid way to calculate the forage feed demand. you need to use eq 10 and mu-forage. The same FCR may be used due to data limitations, but FCR could vary based on forage vs feed (and even the type of feed).

AR: Thank you for your comment. We agree that there is a factor of 1/FE (or FCR) missing here and the text has been updated to reflect this.

lines 278-279: Is this the maximum proportion of each waste type available for a given product? Or the maximum proportion of the product diet filled by the given waste type? It seems like the latter based on the table title in the appendix. You need both of these, which are different, for eq 11, but you use only one.

AR: z is the latter as you suggest, however it is also used for the second purpose in the summation at the end, see the next comment.

lines 288-296: This does not seem correct. First of all, the z multiplied by D should be different than the z used to determine the fraction of available waste for product j in term S. The former is related to how much needed feed comes from waste, while the latter is related to the fraction of each waste stream directed to each product. Second, the first term needs to be removed from the summation and calculated as the minimum feed-from-feed value, using 1-sum(zjw) with zjw being the max proportion of feed coming from waste w. Third, the S term needs to be added to the minimum feed-from-feed value instead of subtracted because the S term calculates the amount of feed energy not available from waste. As it is, you are summing slightly different and slightly adjusted minimum feed-from-feed values to overestimate the final value. Also, G in the text should be omega.

AR: Thank you for these comments, Equation 11 was structured incorrectly as you point out. These ratios are deliberately the same, the final Z / sum(Z) is the fraction used to direct waste to each product, I was unable to find data regarding how waste is distributed among livestock groups. However, since these ratios convey a 'propensity' of the livestock to the consumption of the waste stream I believe this makes sense. The second and third points you make have been addressed as you suggest.

**Land use**

It seems like you would need to calculate the food crop area first, then the waste component of feed because you don't know harvest residue availability until crop area is determined, then fodder component of feed to get fodder crop area, and then finally pasture area. Is this correct?

AR: This is correct, the order of calculations in the model itself is different to the order in which they are discussed here.

lines 351-352: Why not just calculate the yield trajectory from the yield data? It seems like an unnecessary step to do the feed calcs and subtraction and then calibrate to the yield value.

AR: Thank you for your comment, the text here was not clear and has been updated. The reason for this is the behaviour described in Appendix F. The calibration factor is based on the anchor rather than being scenario specific, but the subsequent land-use scaling is scenario specific to address minor discrepancies at the boundary between historic and modelled in non-anchor scenarios.

**Anchor scenario**

line 376: the modelled reversal in Europe pasture needs to be explained. if this is based on historical data/trends, then pasture should not be increasing after it has been decreasing historically.

AR: This is a result of the trade mechanic, almost all of this pasture expansion occurs in Russia, which ranks 4th for global beef production and consistently in the top 6 for other animal products (I have traced the pasture area back through the model; the increase in area comes from an increase in production demand, a direct result of the trade mechanism). We appreciate that the trade method is a significant limitation of the model but don't believe that it undermines the efficacy of the model for broad scale sensitivity experiments. I have added a sentence to explain this.

**Sensitivity**

line 442: Isn't the anchor projected diet based on historical trends? Then it isn't idealized.

AR: thank you for your comment, this sentence was a mistake left over from the previous iteration and has been edited to convey the increasing calorie intake (from the projection rather than prescription).

line 450: do you mean "setting" rather than "halting"?

AR: setting is a better word here, thank you for this suggestion.

**Discussion**

I suggest you tie your sensitivity and vegetarian example to the potential uses of the model and what we can learn from it. For example, show how different land use types change under the vegetarian scenario and discuss the implications.

*AR:* Thank you for your comment, some discussion of the validity of the vegetarian scenario and potential further uses of the model has been added.

**Figure A2**

n number of countries does not add up

*AR*: Thanks for your comment, the numbers in the caption were incorrect, the figure numbers are correct (I had failed to exclude states with zero / NaN production in the caption).